# GaussMarker: Robust Dual-Domain Watermark for Diffusion Models

Kecen Li [* 1 2]  Zhicong Huang [3]  Xinwen Hou [1]  Cheng Hong [3]

## Abstract

As Diffusion Models (DM) generate increasingly realistic images, related issues such as copyright and misuse have become a growing concern. Watermarking is one of the promising solutions. Existing methods inject the watermark into the *single-domain* of initial Gaussian noise for generation, which suffers from unsatisfactory robustness. This paper presents the first *dual-domain* DM watermarking approach using a pipelined injector to consistently embed watermarks in both the spatial and frequency domains. To further boost robustness against certain image manipulations and advanced attacks, we introduce a model-independent learnable Gaussian Noise Restorer (GNR) to refine Gaussian noise extracted from manipulated images and enhance detection robustness by integrating the detection scores of both watermarks. GaussMarker efficiently achieves state-of-the-art performance under eight image distortions and four advanced attacks across three versions of Stable Diffusion with better recall and lower false positive rates, as preferred in real applications.

## 1. Introduction

Diffusion models (Ho et al., 2020; Song et al., 2021; Rombach et al., 2022) have marked a significant advancement in image generation, empowering individuals from various backgrounds to effortlessly create high-quality images. However, these highly realistic images can be misused in several ways, such as faking synthetic images as human-created ones, generating fake news, and copyright infringement (Humphreys et al., 2024).

To mitigate these concerns, watermarking has become a

*Work done during an internship at Ant Group. [1]Institute of Automation, Chinese Academy of Sciences [2]School of Artificial Intelligence, University of Chinese Academy of Sciences [3]Ant Group. Correspondence to: Zhicong Huang <zhicong.hzc@antgroup.com>.

*Proceedings of the 42$^{nd}$ International Conference on Machine Learning*, Vancouver, Canada. PMLR 267, 2025. Copyright 2025 by the author(s).

promising solution, and governments are starting to exert pressure on companies to adopt watermarks (Biden, 2023; California State Legislature, 2024; European Union, 2024). A watermarking method usually consists of two modules: a watermark injector and a watermark detector. The injector embeds a watermark into our target (e.g., a synthetic image), while the detector is capable of extracting the corresponding watermark from the watermarked target. To watermark a diffusion model, a naive approach is to apply classical image watermarking techniques (Cox et al., 2007) to the contents generated by the diffusion model. These traditional watermarks are easy to remove without compromising content quality and lack adequate robustness for detection in real applications (Ren et al., 2024). Therefore, some researchers propose to inject the watermark into the parameters of diffusion model (Fernandez et al., 2023; Cui et al., 2023; Xiong et al., 2023; Zhao et al., 2023; Rezaei et al., 2024; Feng et al., 2024). With the parameters watermarked, the images generated by the diffusion model also contain some information for the detector to extract the watermark. However, these approaches need to fine-tune the diffusion model, inevitably introducing additional computational overhead.

Recent research delves into designing tuning-free watermarking approaches (Gunn et al., 2024; Yang et al., 2024; Ci et al., 2024; Wen et al., 2023; Wang et al., 2024). Since the diffusion model generates images through denoising a Gaussian noise iteratively, they use a carefully designed injector to embed the watermark into the initial Gaussian noise. For detection, they leverage the Denoising Diffusion Implicit Model (DDIM) inversion (Song et al., 2021) to estimate the initial noise from the input image and detect whether the watermark exists in the estimated Gaussian noise. Although these tuning-free approaches can resist some attack algorithms by leveraging the inherent generalization of diffusion models, they still suffer from detection performance degrading under some easy image editing. For example, when the watermarked image is rotated by just 3°, the watermark detection accuracy of Gaussian Shading (Yang et al., 2024) (the SOTA tuning-free method) decreases from 100% to 64%, making this method extremely vulnerable in practical application scenarios.

This paper proposes **GaussMarker**, a more robust tuning-free method to watermark diffusion models. Compared to existing methods which can only watermark the Gaus-

sian noise a single time (in the spatial domain or frequency domain), we design a pipelined injector to *consistently* embed watermarks across both the spatial and frequency domains, thereby implementing a dual-domain watermarking approach. The inspiration comes from the success of dual-domain watermark in traditional image watermarking (Shih & Wu, 2003). We discover that while this strategy results in better performance compared to single-domain approaches, its robustness is still inadequate, particularly in the face of rotation, cropping, and several more advanced attacks. To address this limitation, we propose a learnable Gaussian Noise Restorer (GNR), which is capable of restoring Gaussian noise to its original state even after significant manipulations of the watermarked image. The GNR is model-independent and does not rely on the diffusion models being watermarked. Ultimately, we enhance detection robustness by fusing the detection scores of both watermarks.

We summarize the contributions of this work as follows:

1. We design the first dual-domain watermarking, GaussMarker, for diffusion models. GaussMarker does not need to fine-tune the diffusion models, while exhibiting strong robustness.

2. We propose a learnable Gaussian Noise Restorer (GNR). GNR does not rely on some diffusion model for training, while significantly enhancing the robustness of watermarking under rotation and cropping attack.

3. Thorough experiments show that, on three stable diffusion models and eight image distortions, the average true positive rate and bit accuracy of GaussMarker surpasses existing methods, validating the superiority of GaussMarker in watermarking diffusion models.

## 2. Related Works

### 2.1. Diffusion Models

Diffusion Models (DM) (Ho et al., 2020; Song & Ermon, 2019) are a class of score-based generative models that learn to reverse a process that gradually degrades the training data structure. For generation, diffusion models sample new images by iteratively denoising an initial noise map into a clean image. DM is the currently strongest model for image generation (Dhariwal & Nichol, 2021), but usually needs much more time for generation compared to the classical generative model GAN, especially for high-resolution image generation.

In order to accelerate the practical usage, Latent Diffusion Models (LDM) (Rombach et al., 2022) is proposed. LDM first trains a Variational AutoEncoder (VAE) (Kingma & Welling, 2014), consisting of an encoder $\mathcal{E}$ and a decoder $\mathcal{D}$, on the high-resolution images $\in \mathbb{R}^{C \times W \times H}$. With the

encoder $\mathcal{E}$ compressing the high-resolution images into a latent space $\mathcal{Z} = \mathbb{R}^{c \times w \times h}$, LDM trains a DM in the latent space. During inference, LDM first samples an initial noise map $z_T \in \mathbb{R}^{c \times w \times h}$ from a standard Gaussian distribution $\mathcal{N}(0, I)$, where $T$ is the time step of DM. After obtaining the clean latent vector $z_0$ through the denoising process of DM, LDM uses the decoder $\mathcal{D}$ to decode the latent vector $z_0$ into the clean image $x_0$. In this paper, we focus on watermarking LDM due to its powerful generative capabilities and broad applications.

### 2.2. Watermark Diffusion Models

Existing watermarking methods can be categorized into two types: post-processing and in-processing schemes (An et al., 2024). Post-processing methods directly embed the watermark into the generated image, while in-processing methods embed the watermark through modifying the generation process or parameters of diffusion models. Compared to post-processing methods, which have been developed for decades, in-processing methods exhibit more promising capability for undetectable watermark (An et al., 2024; Wan et al., 2022). According to whether the model's parameters are fine-tuned, existing in-processing watermarks can be mainly divided into two types: tuning-based watermarks and tuning-free watermarks. Since GaussMarker belongs to tuning-free watermarks, we put the introduction of tuning-based watermarks into Appendix A.

**Tuning-free Watermarks.** Most existing tuning-free watermarks inject the watermark into the initial Gaussian noise $z_T$ during the generation. For detection, they leverage the Denoising Diffusion Implicit Model (DDIM) inversion (Song et al., 2021) to estimate $z_T$ from the input image, and detect whether the watermark exists. Inspired by that a number of classical watermarking strategies rely on watermarking images in Fourier space (Cox et al., 2007), Tree-Ring (Wen et al., 2023) proposes to inject the watermark into the initial Gaussian noise in the *frequency domain*. Later, RingID (Ci et al., 2024) enhances Tree-Ring with a novel multi-channel heterogeneous watermarking approach for multi-key identification. Benefiting from the invariances of Fourier space, these methods exhibit strong robustness under many image distortions. However, they restrict the generation randomness of LDM and compromise model performance to inject watermarks. To mitigate this, Gaussian Shading (Yang et al., 2024) proposes a performance-lossless watermark. Compared to Tree-Ring, Gaussian Shading injects watermarks into the *spatial domain*. Specifically, they select a fixed quadrant of latent space as the watermarking key and only generate images from Gaussian noise in that quadrant. Based on Gaussian Shading, PRC (Gunn et al., 2024) introduces a pseudorandom code (Christ & Gunn, 2024) to enhance the variability of generated images. Recently, LatentTracer (Wang et al., 2024) find that images generated

by LDM are naturally watermarked by the decoder. They trace the generated images by checking if the images can be well-reconstructed with an inverted latent vector. However, LatentTracer is not robust to many image distortions, limiting its practical applications.

## 3. Method

Compared to tuning-based watermarks, existing tuning-free watermarks exhibit a certain gap in the robustness of watermark detection, limiting their practical applications. This paper proposes GaussMarker, a robust dual-domain watermark with two significant improvements. (1) GaussMarker injects the watermark into the spatial domain and frequency domain of the Gaussian noise simultaneously. During detection, the detection scores from two domains are fused to achieve more robust detection. (2) GaussMarker introduces a Gaussian Noise Restorer (GNR) to enhance its robustness especially under the rotation and cropping attacks. Fig. 1 gives an overview of our methodology.

### 3.1. Watermark Injection

Given a randomly sampled Gaussian noise map $z_T \in \mathbb{R}^{c \times w \times h}$ and a $l$-bit watermark $\omega \in \{0,1\}^l$ ($l < c \times w \times h$), GaussMarker injects $\omega$ into $z_T$ in the spatial and frequency space sequentially.

#### 3.1.1. SPATIAL-DOMAIN INJECTION

GaussMarker performs spatial-domain injection through using $\omega$ to change the signal of $z_T$. To achieve this, we first rescale $\omega$ into a signal map $s \in \{0,1\}^{c \times w \times h}$ which has the same dimension with $z_T$. The rescale rule is expected to have two characteristics: (1) invertible so that we can recover $\omega$ from $s$ during detection, and (2) the rescale result is sufficiently chaotic to ensure the independence of each dimension in $z_T$. In GaussMarker, the rescaling has two processes. First, $\omega$ is up-sampled into the target dimension with the nearest sampling. Second, we use a deterministic code (Bernstein et al., 2008) to perform shuffling. Formally, $s$ is obtained as

$$s = \text{Shuffle} \left( \text{Up-Sample} \left( \omega, c \times w \times h \right), k \right), \quad (1)$$

where $k$ is a key of the code used for shuffling. With $k$, we can decode the code input from $s$. Then, the spatial-watermarked Gaussian noise can be formalized as

$$z_T^s = \text{abs} \left( z_T \right) \cdot \left( 2s - 1 \right), \quad (2)$$

where $\text{abs} \left( z_T \right)$ denotes taking the absolute value of $z_T$. In essence, randomized $s$ is used to instruct the signs of Gaussian noise $z_T$, analogous to the strategy used by PRC-based watermark (Gunn et al., 2024). Since the whole watermark $\omega$ is injected into $z_T$, the watermark capacity in the spatial domain is $l$ bits.

#### 3.1.2. FREQUENCY-DOMAIN INJECTION

GaussMarker implements frequency-domain injection through further modifying the spatial-watermarked Gaussian noise map $z_T^s$. In order to ensure the detectability of spatial-domain watermark, frequency-domain injection is expected to preserve the signal of Gaussian noise map. Intuitively, injecting a watermark of more bits usually requires a larger scale of editing. Therefore, GaussMarker injects a zero-bit watermark into the frequency domain of $z_T^s$.

We first obtain the frequency feature of $z_T^s$ by performing Fourier transformation on $z_T^s$ as

$$\hat{z}_T^s = \mathcal{F} \left( z_T^s \right), \quad (3)$$

where $\mathcal{F}$ denotes the Fourier transform, and $\hat{z}_T^s \in \mathbb{R}^{c \times w \times h}$ is the Fourier feature of $z_T^s$. We pre-define a mask $M \in \{0,1\}^{c \times w \times h}$, representing the location of $\hat{z}_T^s$ to be edited. The dual-domain watermarked Gaussian noise can be formalized as

$$z_T^{s,f} = \mathcal{F}^{-1} \left( \hat{z}_T^s \cdot (1 - M) + \omega^f \cdot M \right), \quad (4)$$

where $\mathcal{F}^{-1}$ denotes inverse Fourier transform, and $\omega^f \in \mathbb{R}^{c \times w \times h}$ is a Fourier feature map which contains a zero-bit watermark. To enhance the robustness of this zero-bit watermark, we construct a ring-like Fourier feature map as (Wen et al., 2023) and adopt a circle mask for $M$ to preserve the ring shape of $\omega^f$. Please refer to Appendix B.2 for more details. In general, a ring with a small radius introduces minimal edits to the space features $z_T^s$ according to Parseval's theorem, essentially preserving the signals $s$. We also explore the effects of mask radius in Appendix D.5.

After obtaining the two-domain watermarked Gaussian noise map $z_T^{s,f}$, the subsequent generation process is the same as the regular one of LDM. Specifically, we can use the UNet $\mathcal{U}$ to iteratively denoise $z_T^{s,f}$ (with a prompt $c$) into the clean $z_0^{s,f}$ with a sampler (Lu et al., 2022; Song et al., 2021). The finial watermarked image is $x^{s,f} = \mathcal{D}(z_0^{s,f})$.

### 3.2. Watermark Detection

Given an image $x$, GaussMarker can detect whether our watermark $\omega$ exists in this image by outputting a detection score $r$. The higher the $r$, the higher the likelihood that $\omega$ exists in $x$.

Similar to existing tuning-free methods, GaussMarker uses DDIM inversion (Song et al., 2021) to estimate a possible original Gaussian noise map $\tilde{z}_T$ of the input image $x$. Specifically, GaussMarker uses the LDM encoder $\mathcal{E}$ to obtain the clean latent $z_0 = \mathcal{E}(x)$. Then, the original Gaussian noise map $\tilde{z}_T$ can be estimated with $z_0, \mathcal{U}$, and an empty prompt through DDIM inversion. The following sections will introduce how GaussMarker extracts the detection score $r_s$

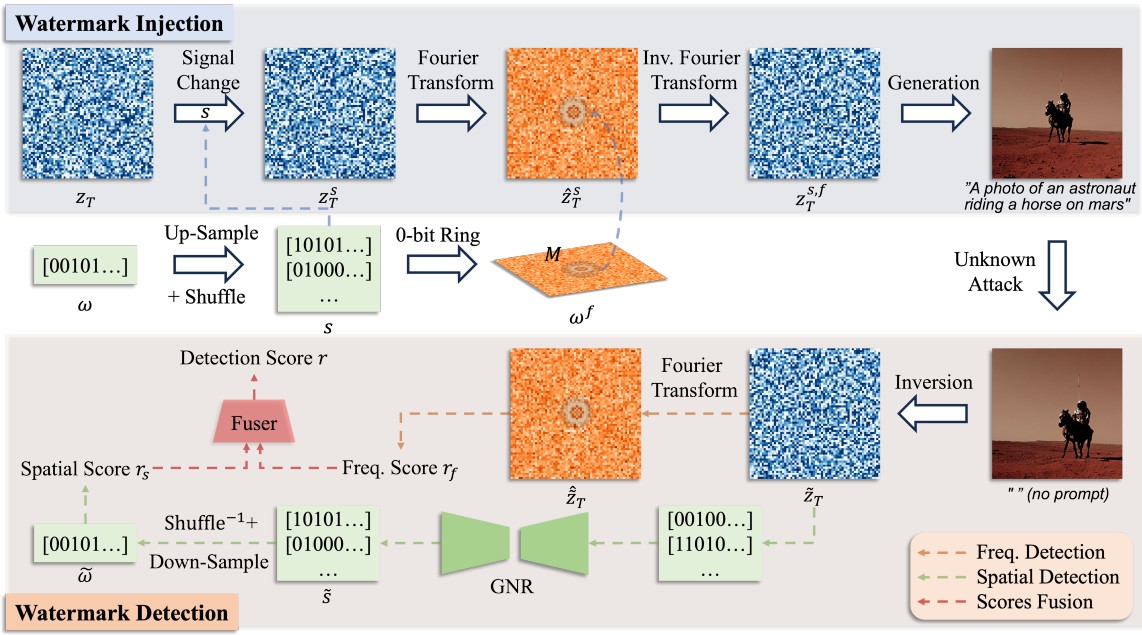

*Figure 1.* Overview of GaussMarker. The $l$-bits watermark $\omega$ is up-sampled and shuffled into a signal map $s$ for $l$-bits spatial-domain watermark. $s$ is used to sample a Fourier map $\omega^f$ for zero-bit frequency-domain watermark. Both $s$ and $\omega^f$ are fixed during the injection and detection. (1) Watermark Injection. We inject a multi-bit watermark and a zero-bit watermark into the spatial domain and frequency domain of Gaussian noise map $z_T$ respectively. (2) Watermark Detection. We extract the frequency and spatial detection scores simultaneously with GNR enhancing the robustness, and fuse two scores to make more robust detection.

and $r_f$ from the spatial and frequency domains of $\tilde{z}_T$ respectively and enhances robustness through GNR and score fusion.

### 3.2.1. SINGLE-DOMAIN DETECTION SCORE

**Spatial-Domain.** GaussMarker extracts the watermark $\tilde{\omega}$ from $\tilde{z}_T$ through the reverse process of spatial-domain injection. Specifically, we first obtain its signal map $\tilde{s} \in \{0,1\}^{c \times w \times h}$ by associating 0 with a negative noise and 1 with a positive noise. Then, $\tilde{\omega}$ can be obtained through a reverse process of Eq.(1) as

$$\tilde{\omega} = \text{Down-Sample}\left(\text{Shuffle}^{-1}\left(\tilde{s}, k\right), l\right), \quad (5)$$

where Down-Sample() uses average pooling. Since Up-Sample() in Eq.(1) uses nearest sampling, Down-Sample() in Eq.(5) acts like a voting strategy. For example, if we inject a 1-bit $\omega = \{0\}$, we up-sample $\omega$ into $s = \{0,0,0,0\}$ using nearest sampling. During detection, if the signal map is estimated as $\tilde{s} = \{0,0,1,0\}$, we down-sample $\tilde{s}$ into the estimated watermark $\tilde{\omega} = \{0.25\}$. The detection score of spatial-domain can be formalized as

$$r_s = -\|\tilde{\omega} - \omega\|^2. \quad (6)$$

**Frequency-Domain.** Similarly, GaussMarker extracts the detection score $r_f$ from $\tilde{z}_T$ through the reverse process of

frequency-domain injection. We first obtain its frequency feature through inverse Fourier transform as

$$\hat{\tilde{z}}_T = \mathcal{F}\left(\tilde{z}_T\right). \quad (7)$$

With pre-defined mask $M$, the detection score is formalized as

$$r_f = -\left\|\left(\hat{\tilde{z}}_T - \omega^f\right) \cdot M\right\|^2. \quad (8)$$

GaussMarker fuses the two detection scores $r_s$ and $r_f$ to make more robust detection. According to ensemble theory (Wood et al., 2023), the effectiveness of score fusing relies on each score having a relatively good detection performance. However, our findings indicate that both scores perform inadequately when subjected to rotation and cropping attacks, particularly when aiming to maintain a low expected false positive rate. This suggests that GaussMarker is unable to derive a meaningful detection score, even when employing an advanced fusion strategy. To solve this, we introduce a learnable Gaussian Noise Restorer (GNR) to enhance the robustness of GaussMarker.

### 3.2.2. GAUSSIAN NOISE RESTORER (GNR)

Our GNR aims to improve the invariance of estimating $\hat{z}_T$ when the watermarked image undergoes rotation and cropping attacks. Formally, given the function space of

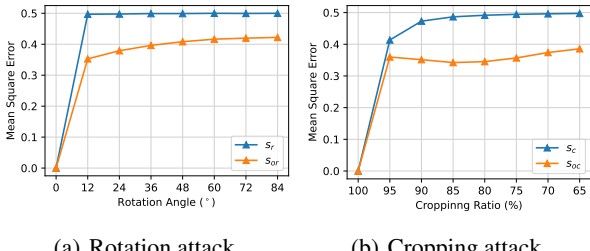

(a) Rotation attack.  (b) Cropping attack.

*Figure 2.* The Mean Square Error between the signal maps estimated from the clean image and image which is edited with rotation ($s_r$) or cropping ($s_c$). $s_{or}$ and $s_{oc}$ are obtained through performing an inverse edition on $s_r$ and $s_c$ respectively. For example, if $s_r$ is estimated from an image that has been rotated by $12°$, we rotate $s_r$ by $-12°$ to obtain $s_{or}$.

GNR $\mathcal{G}$, the objective of GNR can be formalized as

$$\underset{\text{GNR} \in \mathcal{G}}{\text{Minimize}} \left\| \text{GNR} \left( \text{Inversion} \left( \mathcal{T} \left( x^{s,f} \right) \right) \right) - z_T^{s,f} \right\|^2, \quad (9)$$

where Inversion() is the process of estimating the Gaussian noise map from an image, $x^{s,f}$ is generated from the watermarked Gaussian noise map $z_T^{s,f}$, and $\mathcal{T}$ is some image transformation, e.g. rotating $75°$. However, directly optimizing Eq.(9) is infeasible for two reasons. First, it needs LDM to generate training data, which is time-consuming. For example, if the number of steps during both the generation and inversion is 50, to generate each training data for Eq.(9), we need the LDM to forward 100 times. Second, GNR from Eq.(9) is related to the LDM used in training. If we have multiple LDMs to watermark, we need to train multiple GNRs, which brings more time cost. To mitigate this, we can approximate the objective as

$$\underset{\text{GNR} \in \mathcal{G}}{\text{Minimize}} \left\| \text{GNR} \left( \mathcal{T} \left( z_T^{s,f} \right) \right) - z_T^{s,f} \right\|^2. \quad (10)$$

Compared to Eq.(9), Eq.(10) replaces Inversion($\mathcal{T}(x^{s,f})$) with $\mathcal{T}(z_T^{s,f})$, and its optimization does not need any LDM. We just need to randomly generate a watermarked Gaussian noise map $z_T^{s,f}$ and edit it with $\mathcal{T}$. The assumption on which Eq.(10) works is that Inversion($\mathcal{T}(x^{s,f})$) $\approx \mathcal{T}(z_T^{s,f})$. This means that if an image is rotated by $75°$, its estimated Gaussian noise map will also be rotated by approximately $75°$. This relationship hardly exists for many convolution layers in the encoder and UNet. However, a similar relationship exists in their signal maps. As shown in Fig. 2, when the input image is rotated or cropped, the MSE between the original noise signal $s$ and new noise signal $s_r$ or $s_c$ changes dramatically. However, when we perform an inverse transformation on the new signal map, e.g. rotating by the same angle but in the opposite direction, the MSE decreases. This invariant property between the image and Gaussian noise map in the DDIM inversion is also verified by (Wen et al.,

2023). Therefore, the objective of GNR on the signal map is formalized as

$$\underset{\text{GNR} \in \mathcal{G}}{\text{Minimize}} \left\| \text{GNR} \left( \mathcal{T} \left( s_T^{s,f} \right) \right) - s_T^{s,f} \right\|^2,$$

where $s_T^{s,f}$ is the signal map of the watermarked Gaussian noise map. We make two further improvements to this objective. First, since the target $s_T^{s,f}$ belongs to $\{0,1\}^{c \times w \times h}$, we replace Mean Square Error loss with Binary Cross Entropy loss. Second, GNR learned from this objective will have a high false positive rate. This is because a shortcut for this objective is always outputting $s_T^{s,f}$ even when the watermark does not exist. Therefore, we introduce negative sample learning to the objective. The final objective is formalized as

$$\underset{\text{GNR} \in \mathcal{G}}{\text{Maximize}} \left\{ \begin{array}{c} (1 - s_T^{s,f}) \log \left( 1 - \text{GNR} \left( \mathcal{T} \left( s_T^{s,f} \right) \right) \right) + \\ s_T^{s,f} \log \left( \text{GNR} \left( \mathcal{T} \left( s_T^{s,f} \right) \right) \right) + \\ (1 - \mathcal{T} (s_T)) \log \left( 1 - \text{GNR} \left( \mathcal{T} (s_T) \right) \right) + \\ \mathcal{T} (s_T) \log \left( \text{GNR} \left( \mathcal{T} (s_T) \right) \right) \end{array} \right\}, \quad (11)$$

where $s_T$ denotes the signal map of Gaussian noise without a watermark. Eq.(11) means that, for the signal map $s_T^{s,f}$ with a watermark, GNR will output $s_T^{s,f}$ even under some transformation $\mathcal{T}$. While for the signal map $s_T$ without a watermark, GNR will just output the input. With GNR, estimating the watermark $\tilde{\omega}$ can be formalized as

$$\tilde{\omega} = \text{Down-Sample} \left( \text{Shuffle}^{-1} \left( \text{GNR} \left( \tilde{s} \right), k \right), l \right). \quad (12)$$

A limitation of Eq.(11) is that GNR only learns the invariance on the signal map, and can only benefit $r_s$. However, since GaussMarker will fuse two scores to make detection, GNR also benefits the final detection score of GaussMarker.

In our experiments, GNR is implemented with a UNet and the last layer of GNR is the Sigmoid activation function, which ensures the output belongs to $(0, 1)$. During the inference, we use 0.5 as the threshold to discretize the output of GNR to ensure the restored signal map belongs to $\{0, 1\}$.

### 3.2.3. SCORE FUSION

Given the detection score $r_s$ and $r_f$ from spatial domain and frequency domain respectively, GaussMarker fuses two scores into one score, which can be used to make more robust detection. This objective can be formalized as

$$\underset{\text{Fuser} \in \mathcal{F}_u}{\text{Maximize}} \left\{ \begin{array}{c} (1 - y) \log \left( 1 - \text{Fuser} \left( r_s, r_f \right) \right) + \\ y \log \left( \text{Fuser} \left( r_s, r_f \right) \right) \end{array} \right\}, \quad (13)$$

where $\mathcal{F}_u$ is the function space of Fuser(), and $y \in \{0, 1\}$ indicates whether the input image has our watermark. In GaussMarker, we implement Fuser() as a two-layer

*Table 1.* Comparison results of TPR@1%FPR / Bit Accuracy under various image distortions.

| Methods | DM | Clean | Rotate | JPEG | C&S | R. Drop | Blur | S. Noise | G. Noise | Bright | Average |
|---|---|---|---|---|---|---|---|---|---|---|---|
| DwtDctSvd | | 0.000 / 0.999 | 0.000 / 0.915 | 0.000 / 0.608 | 0.000 / 0.839 | 0.000 / 0.991 | 0.000 / 0.978 | 0.028 / 0.422 | 0.614 / 0.759 | 0.000 / 0.756 | 0.071 / 0.807 |
| Stable Signature | | **1.000** / 0.992 | 0.957 / 0.859 | 0.926 / 0.802 | **1.000** / 0.978 | 0.983 / 0.912 | 0.949 / 0.752 | 0.918 / 0.812 | 0.948 / 0.880 | 0.921 / 0.861 | 0.956 / 0.872 |
| Tree-Ring | SD V2.1 | **1.000** / - | 0.548 / - | 0.998 / - | 0.048 / - | 0.994 / - | **1.000** / - | 0.979 / - | 0.913 / - | 0.927 / - | 0.823 / - |
| RingID | | **1.000** / - | **1.000** / - | **1.000** / - | 0.078 / - | **1.000** / - | **1.000** / - | **1.000** / - | 0.944 / - | 0.928 / - | 0.883 / - |
| Gaussian Shading | | **1.000** / **1.000** | 0.018 / 0.512 | 0.999 / 0.986 | 0.081 / 0.540 | **1.000** / **0.964** | **1.000** / 0.999 | 0.999 / 0.923 | **0.996** / 0.941 | **0.998** / **0.998** | 0.788 / 0.874 |
| PRC | | **1.000** / **1.000** | 0.010 / 0.500 | 0.926 / 0.962 | 0.020 / 0.501 | 0.957 / 0.978 | 0.993 / 0.996 | 0.392 / 0.692 | 0.836 / 0.917 | 0.606 / 0.800 | 0.638 / 0.816 |
| LatentTracer | | 0.990 / - | 0.017 / - | 0.010 / - | 0.013 / - | 0.088 / - | 0.012 / - | 0.010 / - | 0.012 / - | 0.059 / - | 0.135 / - |
| **GaussMarker** | | **1.000** / **1.000** | 0.997 / **0.998** | 0.996 / **0.997** | **1.000** / **1.000** | **1.000** / 0.963 | **1.000** / **1.000** | 0.999 / **0.991** | 0.989 / **0.968** | 0.993 / 0.989 | **0.997** / **0.990** |
| DwtDctSvd | | 0.000 / 0.999 | 0.000 / 0.902 | 0.000 / 0.610 | 0.000 / 0.819 | 0.000 / **0.991** | 0.000 / 0.971 | 0.035 / 0.419 | 0.574 / 0.724 | 0.000 / 0.741 | 0.068 / 0.797 |
| Stable Signature | | **1.000** / 0.991 | 0.940 / 0.860 | 0.907 / 0.809 | **1.000** / 0.978 | 0.978 / 0.914 | 0.949 / 0.754 | 0.928 / 0.815 | 0.915 / 0.867 | 0.938 / 0.867 | 0.950 / 0.873 |
| Tree-Ring | SD V2.0 | **1.000** / - | 0.552 / - | **1.000** / - | 0.051 / - | 0.997 / - | **1.000** / - | 0.981 / - | 0.897 / - | 0.948 / - | 0.822 / - |
| RingID | | **1.000** / - | **1.000** / - | **1.000** / - | 0.116 / - | **1.000** / - | **1.000** / - | **1.000** / - | 0.953 / - | 0.995 / - | 0.896 / - |
| Gaussian Shading | | **1.000** / **1.000** | 0.001 / 0.514 | 0.998 / 0.988 | 0.171 / 0.543 | **1.000** / 0.969 | **1.000** / 0.999 | 0.999 / 0.932 | **0.998** / 0.944 | **0.999** / 0.976 | 0.796 / 0.874 |
| PRC | | **1.000** / **1.000** | 0.016 / 0.502 | 0.826 / 0.912 | 0.009 / 0.500 | 0.802 / 0.900 | 0.980 / 0.989 | 0.257 / 0.624 | 0.363 / 0.676 | 0.720 / 0.858 | 0.552 / 0.773 |
| LatentTracer | | 0.994 / - | 0.014 / - | 0.011 / - | 0.007 / - | 0.022 / - | 0.016 / - | 0.019 / - | 0.029 / - | 0.003 / - | 0.124 / - |
| **GaussMarker** | | **1.000** / **1.000** | 0.997 / **0.999** | 0.998 / **0.998** | **1.000** / **1.000** | **1.000** / 0.969 | **1.000** / **1.000** | 0.996 / **0.992** | 0.993 / **0.970** | 0.998 / **0.993** | **0.998** / **0.991** |
| DwtDctSvd | | 0.000 / 0.999 | 0.000 / 0.915 | 0.000 / 0.619 | 0.048 / 0.832 | 0.000 / **0.991** | 0.000 / 0.975 | 0.041 / 0.418 | 0.738 / 0.745 | 0.000 / 0.743 | 0.092 / 0.804 |
| Stable Signature | | 0.999 / 0.993 | 0.957 / 0.881 | 0.932 / 0.805 | 0.998 / 0.984 | 0.977 / 0.925 | 0.903 / 0.749 | 0.912 / 0.829 | 0.940 / 0.900 | 0.931 / 0.879 | 0.950 / 0.883 |
| Tree-Ring | SD V1.4 | **1.000** / - | 0.565 / - | **0.999** / - | 0.031 / - | **1.000** / - | **1.000** / - | 0.981 / - | 0.930 / - | 0.942 / - | 0.828 / - |
| RingID | | 0.950 / - | 0.950 / - | 0.950 / - | 0.116 / - | 0.948 / - | 0.950 / - | 0.568 / - | 0.634 / - | 0.945 / - | 0.779 / - |
| Gaussian Shading | | **1.000** / **1.000** | 0.011 / 0.516 | **0.999** / 0.989 | 0.096 / 0.542 | **1.000** / 0.971 | **1.000** / 0.999 | **1.000** / 0.923 | **0.998** / 0.943 | **0.998** / 0.979 | 0.789 / 0.874 |
| PRC | | 0.999 / 0.999 | 0.004 / 0.500 | 0.831 / 0.915 | 0.012 / 0.500 | 0.865 / 0.932 | 0.969 / 0.985 | 0.245 / 0.621 | 0.374 / 0.683 | 0.735 / 0.867 | 0.559 / 0.778 |
| LatentTracer | | 0.000 / - | 0.000 / - | 0.000 / - | 0.000 / - | 0.000 / - | 0.000 / - | 0.000 / - | 0.000 / - | 0.000 / - | 0.000 / - |
| **GaussMarker** | | **1.000** / **1.000** | **0.998** / **0.999** | 0.998 / **0.998** | **0.999** / **0.999** | **1.000** / 0.970 | **1.000** / **1.000** | **1.000** / **0.991** | 0.988 / **0.965** | 0.996 / **0.994** | **0.998** / **0.991** |

*Table 2.* Comparison results of CLIP Score and FID across three Stable Diffusion V1.4 / V2.0 / V2.1.

| Methods | CLIP Score ↑ | Ave. | FID ↓ | Ave. |
|---|---|---|---|---|
| No Watermark | **0.3488** / 0.3580 / 0.3633 | 0.3567 | 25.05 / 24.40 / 25.22 | 24.89 |
| DwtDctSvd | 0.3424 / 0.3516 / 0.3580 | 0.3507 | 24.47 / 23.74 / **24.21** | **24.14** |
| Stable Signature | 0.3466 / 0.3545 / 0.3611 | 0.3541 | 24.23 / 24.55 / 25.02 | 24.60 |
| Tree-Ring | 0.3468 / 0.3586 / 0.3644 | 0.3566 | 24.86 / 24.94 / 25.59 | 25.13 |
| RingID | 0.3282 / 0.3534 / 0.3603 | 0.3473 | 27.27 / 25.04 / 26.17 | 26.16 |
| Gaussian Shading | 0.3467 / **0.3591** / **0.3646** | **0.3568** | 24.64 / **23.67** / 24.77 | 24.36 |
| PRC | 0.3466 / 0.3576 / 0.3482 | 0.3508 | **24.19** / 24.30 / 24.80 | 24.43 |
| LatentTracer | 0.3488 / 0.3580 / 0.3633 | 0.3567 | 25.05 / 24.40 / 25.22 | 24.89 |
| **GaussMarker** | 0.3427 / 0.3578 / 0.3631 | 0.3545 | 25.00 / 24.44 / 25.11 | 24.85 |

MLP and generate 100 watermarked images and 100 un-watermarked images for training. During inference, the final detection score of GaussMarker is $r = \text{Fuser}(r_s, r_f)$.

## 4. Experiments

### 4.1. Experimental Setup

**Baselines.** We mainly compare our GaussMarker with tuning-free methods, including Tree-Ring (Wen et al., 2023), RingID (Ci et al., 2024), Gaussian Shading (Yang et al., 2024), PRC (Gunn et al., 2024), and LatentTracer (Wang et al., 2024). Additionally, we also select a tuning-based method, Stable Signature (Fernandez et al., 2023) and a classic image watermarking method, DwtDctSvd (Cox et al., 2007), which is officially used by Stable diffusion. Each baseline generates 1,000 watermarked images and 1,000 un-watermarked images for evaluation. Since LatentTracer detects synthetic images without artificial watermark, the un-watermarked images are generated by a LDM different from the LDM to be watermarked. For example, if the watermarked images are generated by SD V1.4, the un-

watermarked images are generated by SD V2.1. All these baselines are implemented with their source code.

**GaussMarker.** For DDIM inversion, we use a guidance scale of 0, 50 inversion steps and, an empty prompt. For spatial watermark, we use a random 256-bit watermark and take a secure stream cipher, ChaCha20 (Bernstein et al., 2008), as the Shuffle. We implement GNR as a 30M UNet and train it with a learning rate of 0.0001, batch size of 32, and training steps of 50,000. The image transformation $\mathcal{T}$ includes the random rotation between -180 and 180 degrees, the random cropping between 70% and 100% and the random sign flipping ($p = 0.35$). We train the Fuser with a learning rate of 0.001, batch size of 200, and training steps of 1,000. More implementation details can be found in Appendix B.

**Diffusion Models.** We focus on text-to-image latent diffusion model. Following previous works (Gunn et al., 2024; Yang et al., 2024), we evaluate our method and other baselines on Stable Diffusion (SD) V2.1, SD V2.0, and SD V1.4. All these models generate 512×512 images with 4×64×64 latent space. For generation, we use the prompts from Stable-Diffusion-Prompts with a guidance scale of 7.5 and 50 sampling steps[1].

**Datasets and Metrics.** We use two metrics to evaluate the detection performance. We calculate the true positive rate (TPR) corresponding to the fixed false positive rate (FPR) 0.01, named TPR@1%FPR. We also calculate the bit accuracy for those multi-bit watermarking methods. For the visual metrics, we compute the FID (Heusel et al., 2017)

---

[1] https://huggingface.co/datasets/Gustavosta/Stable-Diffusion-Prompts

and CLIP-Score (Radford et al., 2021). The FID is calculated using 5000 prompt and image pairs sampled from MS-COCO (Lin et al., 2014).

## 4.2. Comparison Results

**Robustness.** We compare GaussMarker with seven baselines under eight image distortions across SD v2.1, SD v2.0 and SD v1.4. The examples of distortions are present in Fig. 4. As shown in Tab. 1, GaussMarker exhibits strong robustness on both TPR1%FPR and bit accuracy. Specifically, GaussMarker surpasses the best-performing baseline, Stable Signature, by 4.6% and 11.5% in terms of the average TPR1%FPR and bit accuracy respectively. These improvements are achieved despite Stable Signature being extensively trained. Among all the tuning-free baselines, RingID obtains the best TPR1%FPR on SD V2.1, but its performance is not consistent on other SDs and is much worse than ours. Surprisingly, on SD V1.4, LatentTracer can not obtain useful TPR1%FPR even when the watermarked image is not edited. We consider the reason is that since LatentTracer uses the reconstruction loss to detect whether an LDM generates the given image, if the image is generated by a less strong LDM (SD V1.4), it can still obtain a small reconstruction loss on a stronger LDM (SD V2.1). Therefore, LatentTracer easily thinks that this image is also generated by this powerful LDM, which causes a high FPR. Among all the methods evaluated, GaussMarker is the sole approach that consistently achieves an average TPR1%FPR and bit accuracy exceeding 0.99, significantly outperforming the other methods.

**Fidelity.** Tab. 2 presents the CLIP score and FID comparison results. For both two metrics, GaussMarker outperforms Stable Signature, the best-performing baseline in robustness comparison. Although Gaussian Shading obtains the best CLIP score, GaussMarker is only 0.6% behind while obtaining 11.7% higher average bit accuracy and 20.7% higher TPR1%FPR, which means that GaussMarker achieves a better trade-off between the model utility and the robustness of detection. For FID, the post-processing method DwtDctSvd performs the best, but it can not obtain a useful detection accuracy as shown in Tab. 1.

## 4.3. Advanced Attacks

This section evaluates the robustness of GaussMarker under more advanced attacks. According to the comparison results in Sec. 4.2, we select the top-4 methods as the baselines in terms of the average TPR1%FPR. We consider three types of attacks as follows.

**Compression Attack.** Following previous works (Yang et al., 2024), we utilize two pre-trained VAE compressors, termed as VAE-1 (Cheng et al., 2020) and VAE-2 (Ballé et al., 2018) respectively, for image compression.

*Table 3.* Comparison results of TPR@1%FPR / Bit Accuracy under four advanced attacks.

| Methods | DM | VAE-1 | VAE-2 | Diffusion | UnMarker |
|---|---|---|---|---|---|
| Stable Signature | | 0.312 / 0.687 | 0.322 / 0.682 | 0.006 / 0.535 | 1.000 / 0.958 |
| Tree-Ring | | 0.955 / - | 0.984 / - | 0.125 / - | 0.011 / - |
| RingID | SD V2.1 | 1.000 / - | 1.000 / - | 0.194 / - | 0.044 / - |
| Gaussian Shading | | 0.994 / 0.982 | 1.000 / 0.982 | 0.066 / 0.530 | 0.081 / 0.544 |
| **GaussMarker** | | 1.000 / 0.999 | 1.000 / 1.000 | 0.667 / 0.865 | 1.000 / 0.994 |
| Stable Signature | | 0.510 / 0.697 | 0.527 / 0.697 | 0.011 / 0.544 | 1.000 / 0.961 |
| Tree-Ring | | 0.984 / - | 0.992 / - | 0.136 / - | 0.021 / - |
| RingID | SD V2.0 | 1.000 / - | 1.000 / - | 0.247 / - | 0.025 / - |
| Gaussian Shading | | 0.991 / 0.983 | 0.994 / 0.985 | 0.075 / 0.534 | 0.151 / 0.545 |
| **GaussMarker** | | 1.000 / 0.999 | 1.000 / 0.998 | 0.853 / 0.872 | 1.000 / 0.994 |
| Stable Signature | | 0.295 / 0.689 | 0.365 / 0.678 | 0.000 / 0.537 | 1.000 / 0.971 |
| Tree-Ring | | 0.933 / - | 0.990 / - | 0.130 / - | 0.016 / - |
| RingID | SD V1.4 | 1.000 / - | 0.924 / - | 0.131 / - | 0.022 / - |
| Gaussian Shading | | 1.000 / 0.982 | 1.000 / 0.984 | 0.052 / 0.526 | 0.210 / 0.543 |
| **GaussMarker** | | 1.000 / 1.000 | 1.000 / 0.999 | 0.744 / 0.853 | 1.000 / 1.000 |

*Table 4.* Ablation study on different modules of GaussMarker across three SD V1.4 / V2.0 / V 2.1.

| Spatial | Frequency | GNR | TPR1%FPR | Bit Acc. |
|---|---|---|---|---|
| ✓ | | | 0.786 / 0.794 / 0.791 | 0.871 / 0.872 / 0.868 |
| | ✓ | | 0.777 / 0.794 / 0.770 | - |
| ✓ | ✓ | | 0.943 / 0.956 / 0.946 | 0.871 / 0.872 / 0.868 |
| ✓ | | ✓ | 0.996 / 0.984 / 0.875 | 0.991 / 0.991 / 0.990 |
| ✓ | ✓ | ✓ | 0.998 / 0.998 / 0.997 | 0.991 / 0.991 / 0.990 |

**Regeneration Attack.** Following a recent benchmark (An et al., 2024), we utilize a diffusion model (Dhariwal & Nichol, 2021) pre-trained on ImageNet to perform regeneration attack. Specifically, the watermarked image will undergo multiple cycles of noising and be denoised through this pre-trained diffusion model for regeneration.

**Visually-Aware Attack.** Recently, Kassis et al. proposed the first practical universal attack, UnMarker (Kassis & Hengartner, 2024). They introduce a visually-aware filter to attack semantic watermarks, such as Tree-Ring and Gaussian Shading. Semantic watermarks only preserve the semantics of images, without regard for changes at the pixel level. GaussMarker also belongs to semantic watermarks.

As shown in Tab. 3, GaussMarker still exhibits the best robustness under such four advanced attacks. Although Stable Signature outperforms all tuning-free baselines under various image distortions (Tab. 1), it loses its superiority under advanced compression attacks. All tuning-free methods successfully resist this compression attack, but fail to work under the regeneration attack and UnMarker. Among all these advanced attacks, the regeneration attack using diffusion models degrades the performance of all methods a lot, including GaussMarker, which is consistent with the results in (An et al., 2024). However, GaussMarker still surpasses other baselines a lot.

## 4.4. Ablation Studies

**Dual-Domain and GNR.** Tab. 4 and 5 present the detection and fidelity performance respectively of five different versions of GaussMarker: (1) only injecting spatial-domain watermark, (2) only injecting frequency-domain watermark,

*Table 5.* Fidelity results on different modules of GaussMarker across three SD V1.4 / V2.0 / V 2.1..

| Spatial | Frequency | CLIP Score ↑ | Ave. | FID ↓ | Ave. |
|---|---|---|---|---|---|
| | | 0.3488 / 0.3580 / 0.3633 | 0.3567 | 25.05 / 24.40 / 25.22 | 24.89 |
| ✓ | | 0.3467 / 0.3591 / 0.3646 | 0.3568 | 24.64 / 23.67 / 24.77 | 24.36 |
| | ✓ | 0.3470 / 0.3595 / 0.3639 | 0.3568 | 24.64 / 24.73 / 24.98 | 24.78 |
| ✓ | ✓ | 0.3427 / 0.3578 / 0.3631 | 0.3545 | 25.00 / 24.44 / 25.11 | 24.85 |

*Table 6.* Effects of transformation $\mathcal{T}$ on the bit accuracy of GaussMarker against Rotate / C&S attacks using SD V2.1.

| Crop Ratio | Rotation Angle | | | |
|---|---|---|---|---|
| | w/o | $(-45°, 45°)$ | $(-90°, 90°)$ | $(-180°, 180°)$ |
| w/o | 0.512 / 0.543 | 0.513 / 0.967 | 1.000 / 0.980 | 0.999 / 0.988 |
| (85%, 100%) | 0.512 / 0.996 | 0.515 / 0.967 | 1.000 / 0.999 | 0.998 / 0.999 |
| (70%, 100%) | 0.512 / 1.000 | 0.514 / 1.000 | 0.998 / 1.000 | 0.998 / 1.000 |
| (55%, 100%) | 0.512 / 1.000 | 0.514 / 1.000 | 0.998 / 1.000 | 0.998 / 1.000 |

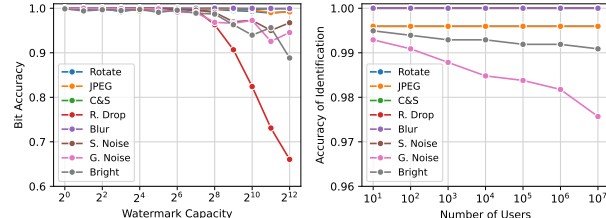

(a) Watermark Capacity. (b) Number of Users.

*Figure 3.* (a) Bit accuracy of GaussMarker when injecting watermark of different bits. (2) Identification accuracy of GaussMarker when allocating watermarks to different numbers of users.

(3) injecting dual-domain watermark without GNR, (4) only injecting spatial-domain watermark and use GNR for detection, and (5) GaussMarker.

For detection, the results show that fusing the detection scores from dual-domain watermarks significantly improves the detection performance. Although GNR can effectively improve the bit accuracy, without score fusion, it could restore some signal maps that do not contain our watermark, increasing the FPR. With both score fusion and GNR, GaussMarker obtains the best TPR1%FPR and bit accuracy. Tab. 8 presents Tab. 4 under various image distortions. We find that spatial-domain watermarks exhibit better robustness due to injecting a watermark of more bits, except for some affine transformations. Fusing integrates the respective advantages of both watermarks, and GNR further enhances the robustness, especially under rotation and cropping.

For fidelity, both spatial and frequency watermarks help preserve the original CLIP Score. For FID, the spatial watermark performs better than the frequency one. Combining both methods often requires more editing, leading to slightly worse CLIP Scores and FID for the dual-domain watermark compared to single-domain approaches. Improving the visual quality of watermarked images using GaussMarker could be a direction for future work.

**Transformation of GNR.** Tab. 6 presents the bit accuracy of GaussMarker under rotation and cropping attacks when training GNR using different scales of rotation and cropping as the transformation $\mathcal{T}$. Note that these two attacks use rotating 75° and cropping 75% respectively. When $\mathcal{T}$ does not include rotating 75°, the bit accuracy under rotation attack is only 51%. When 75° is included, the bit accuracy significantly increases. Surprisingly, even when $\mathcal{T}$ does not include cropping 75%, GaussMarker can also obtain a high bit accuracy, which means that the invariance of cropping can be learned more easily even just through rotation. However, if GNR is trained without any $\mathcal{T}$, the bit accuracy under both attacks is very low.

**Watermark Capacity.** Fig. 3(a) presents the bit accuracy

of GaussMarker when the watermark has different numbers of bits. Since GaussMarker needs the number of bits $l$ to be smaller than the latent size $c \times w \times h$, it can inject a watermark of up to $4 \times 64 \times 64 = 2^{12}$ bits. The results show that, the bit accuracy remains nearly 100% when the number of watermark bits is less than $2^7$ under all image distortions. As the number of bits increases, the bit accuracy starts to decline especially for the Random Drop. This is because each bit in the watermark $\omega$ will be repeated $\frac{c \times w \times h}{l}$ times to perform the nearest up-sampling. During detection, we use average down-sampling to let $\frac{c \times w \times h}{l}$ estimated signal values vote for bit prediction. When $\frac{c \times w \times h}{l}$ is small ($l$ is close to $c \times w \times h$), the benefit of voting naturally diminishes.

**Number of Users.** When we have multiple users using one LDM, the multi-bit watermark is capable of allocating a unique watermark for each user. Given an image, we extract the watermark to identify which user it belongs to. In GaussMarker, we first determine whether the image contains a watermark. If it does, the user with the highest detection score is considered the one who generated the image. In such case, each LDM only has one unique model-watermark $w \in \{0, 1\}^l$ (or signal map) with its corresponding GNR[2]. Each user will be assigned a unique key $k \in \{0, 1\}^l$ and a unique user-watermark $w_u \in \{0, 1\}^l$ with $w_u = \text{XOR}(w, k)$. We use the estimated $\tilde{w}$ to estimate the user-watermark $\tilde{w}_u = \text{XOR}(\tilde{w}, k)$ for calculating the bit accuracy.

Fig. 3(b) presents the identification accuracy of GaussMarker under different numbers of users. Following previous works (Yang et al., 2024; Rezaei et al., 2024), we adopt a more effective approach to calculate the identification accuracy, which is detailed in Appendix C. Even when the number of users is $10^7$, GaussMarker still exhibits almost perfect identification in five cases. Although the identification accuracy for Gaussian Noise is only 97.6%, if a user generates two images, the probability of successfully

---

[2]Therefore, we only need to train one GNR even when we have multiple users.

*Table 7.* The time cost of three modules of GaussMarker during training and detection phase under SD V 2.1.

| Phase | Inversion | GNR | Fuser |
|---|---|---|---|
| Training | - | 72 min | $5.9 \times 10^{-2}$s |
| Detection | 6.5s | $1.2 \times 10^{-3}$s | $1.0 \times 10^{-4}$s |

identifying him is still no less than 99%.

**Time Cost.** Although GaussMarker needs additional training, but the cost is minimal and the training is model-agnostic. As presented in Table 7, the training of GNR only needs 72 minutes on 1 V100 32G GPU. Moreover, the time overhead incurred by GNR and Fuser during the detection phase is almost negligible.

## 5. Limitations

Similar to existing tuning-free DM watermarks, Gauss-Marker also relies on DDIM inversion (Song et al., 2021) to estimate the initial Gaussian map for watermark detection. Therefore, we need to use continuous-time samplers based on ODE solvers (Song et al., 2021). Besides, since the parameters of DM are not watermarked, GaussMarker does not support releasing the model with its parameters.

## 6. Conclusion

This paper proposes GaussMarker, the first dual-domain watermark for diffusion models. Compared to existing single-domain watermarks, we inject the watermark into the spatial and frequency domain of the initial Gaussian noise simultaneously. The detection scores from dual-domain watermarks are fused to achieve a more robust detection. Besides, we introduce GNR which does not rely on any diffusion models for training, while significantly enhancing the robustness under rotation and cropping attacks. Extensive experiments show that GaussMarker exhibits strong robustness and preserves the utility of diffusion models.

## Impact Statement

Studying the security and privacy aspects of the generative models has become a growing concern (Carlini et al., 2023; 2024). This paper proposes a robust DM watermark, which can be used to detect whether an image is generated by a given DM. We believe that our approach can mitigate issues such as copyright infringement and misuse associated with DM, thereby promoting the development of trustworthy generative AI.

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

# A. Tuning-Based Watermarks

(Liu et al., 2023) proposes the first tuning-based method. They construct a dataset containing the pair of watermarked prompts and the watermarked images, as well as the clean prompts and the clean images for fine-tuning LDM. However, they do not provide a clear method to detect whether the watermark exists in an image, and require powerful hardware to fine-tune the models. To solve these problems, Stable Signature proposed to only fine-tune the decoder of LDM (Fernandez et al., 2023). To achieve this, they first train a watermark extractor that can recover the hidden signature from any generated image. Then, they fine-tune the LDM decoder, such that all generated images yield a given signature through the pre-trained extractor. Compared to directly fine-tuning the original decoder, LaWa (Rezaei et al., 2024) proposes to add some intermediate layers into the decoder, and only fine-tune these new layers with other modules frozen. Compared to Stable Signature, LaWa is more robust to image modifications and can handle multiple users for one image generation service. In order to defend against white-box attacks, AquaLoRA proposes to fine-tune the UNet, which contains essential knowledge of LDM (Feng et al., 2024). These methods inevitably introduce additional computational overhead especially when LDM gets large, and their robustness heavily relies on taking sufficient image augmentations during the training.

# B. Implementation Details

## B.1. Baselines

**Stable Signature (Fernandez et al., 2023).** We fine-tune the decoder of SD using the watermark extractor provided in their repo[3]. This extractor model has been trained with more image augmentations, such as blur and rotations, and has better robustness to that kind of attacks, at the cost of a slightly lower image quality. We fine-tune the encoder on a subset of COCO (Lin et al., 2014), which contains 2,000 images, with batch size 4, training steps 100, and leaning rate $5 \times 10^{-4}$.

**LatentTracer (Wang et al., 2024).** LatentTracer considers that the images generated by LDM contain a natural watermark, and does not inject any artificial watermark into the generated image. When detecting the watermark from SD V2.1 and SD V2.0, the un-watermarked images are generated by SD V1.4. When detecting the watermark from SD V1.4, the un-watermarked images are generated by SD V2.1. We use the default hyper-parameters in their repo[4] to obtain the latent construction loss.

[3]https://github.com/facebookresearch/stable_signature

[4]https://github.com/ZhentingWang/LatentTracer

## B.2. GaussMarker

**Up-sample and Down-sample.** Given an $l$-bits watermark $\omega \in \{0,1\}^l$, GaussMarker up-samples it into a signal map $s \in \{0,1\}^{c \times w \times h}$ with nearest sampling. During detection, GaussMarker down-sample the estimated signal map $\tilde{s}$ with average pooling to estimate $\omega$. The mapping relationship of elements in up-sampling and down-sampling needs to be consistent. For example, given $\omega = \{0,1\}$ and $c \times w \times h = 7$, if $s = \{\omega[0], \omega[0], \omega[0], \omega[1], \omega[1], \omega[1], \omega[1]\}$ and $\tilde{s} = \{0,0,1,1,0,1,1\}$, the estimated watermark is $\tilde{\omega} = \{\frac{\tilde{s}[0]+\tilde{s}[1]+\tilde{s}[2]}{3}, \frac{\tilde{s}[3]+\tilde{s}[4]+\tilde{s}[5]+\tilde{s}[6]}{4}\} = \{\frac{1}{3}, \frac{3}{4}\}$. If $s = \{\omega[0], \omega[0], \omega[0], \omega[0], \omega[1], \omega[1], \omega[1]\}$ and $\tilde{s} = \{1,0,0,0,0,1,1\}$, the estimated watermark is $\tilde{\omega} = \{\frac{\tilde{s}[0]+\tilde{s}[1]+\tilde{s}[2]\tilde{s}[3]}{4}, \frac{\tilde{s}[4]+\tilde{s}[5]+\tilde{s}[6]}{3}\} = \{\frac{1}{4}, \frac{2}{3}\}$. This mapping relationship can also be treated as a key and saved into $k$ in the Shuffle.

**Frequency Watermark.** To enhance the robustness of this zero-bit watermark, we construct a ring-like Fourier feature map as (Wen et al., 2023). We first obtain a Gaussian noise map $z_T^f = \text{abs}(\epsilon) \cdot (2s - 1)$, where $\epsilon \sim \mathcal{N}(0, I)$. Then, we calculate its Fourier feature as $\hat{\omega}^f = \mathcal{F}(z_T^f)$ and flatten it into a one-dimensional vector. Formally, the feature map can be defined as

$$\omega^f[:, i, j] = \frac{1}{N_{i,j}} \sum_{\substack{0 \le i^f < w, 0 \le j^f < h, \\ r_{i,j}^2 = r_{i^f, j^f}^2}} \hat{\omega}^f[i^f, j^f], \quad (14)$$

where $i \in \{0, \ldots, w-1\}$, $j \in \{0, \ldots, h-1\}$, $r_{i,j}^2 = \lfloor i - \frac{w}{2}\rfloor^2 + \lfloor j - \frac{h}{2}\rfloor^2$, $r_{i^f, j^f}^2 = \lfloor i^f - \frac{w}{2}\rfloor^2 + \lfloor j^f - \frac{h}{2}\rfloor^2$ and $N_{i,j}$ is the number of summed elements. Eq. 14 implies that, taking the center of $\omega^f$ as the center of a circle, elements at a similar radius will have the same value, thereby forming a ring-like feature map. The value is defined as the average of elements in $\hat{w}^f$ that have the same radius. This characteristic can enhance the robustness through leveraging many properties of the Fourier transform (Wen et al., 2023; Ci et al., 2024). To preserve the ring shape of watermark in $z_T^{s,f}$, we adopt a circle mask for $M$ as

$$M[:, i, j] = \begin{cases} 1, & \text{if } r_{i,j}^2 < r_f^2 \\ 0, & \text{if } r_{i,j}^2 \ge r_f^2 \end{cases}, \quad (15)$$

where $r_f$ is the radius of circle mask. In our experiments, we set $r_f = 4$, and we explore the effects of $r_f$ in Appendix D.5.

## B.3. Image Distortions

As shown in Fig. 4, we adopt eight image distortions to evaluate the robustness of GaussMarker and baselines as follows:

**(b) Rotate.** Rotate the image by 75 degrees and fill the extra areas with black pixels.

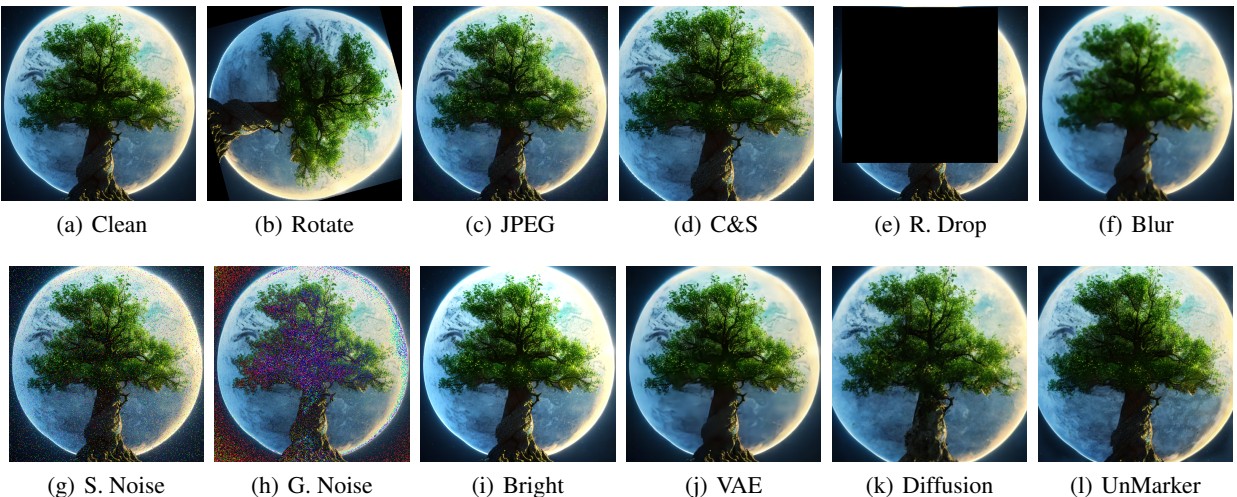

| (a) Clean | (b) Rotate | (c) JPEG | (d) C&S | (e) R. Drop | (f) Blur |

| (g) S. Noise | (h) G. Noise | (i) Bright | (j) VAE | (k) Diffusion | (l) UnMarker |

*Figure 4.* Examples of attacks used in our experiments.

**(c) JPEG.** Compress the image using JPEG and set the quality factor to 25.

**(d) C&S.** Randomly crop 75% area of the image and scale it into the original resolution.

**(e) R. Drop.** Randomly mask 80% area of the image with black pixels.

**(f) Blur.** Apply a median filter with a kernel size of 7 to the image.

**(g) S. Noise.** Add salt-and-pepper noise to the image with a probability of 0.05.

**(h) G. Noise.** Add Gaussian noise with a standard deviation of 0.05 to the image.

**(i) Bright.** Perform a brightness transformation on the image with a factor of 6.

### B.4. Advanced Attacks

**Compression Attack.** Following previous works (Yang et al., 2024), we utilize two pre-trained VAE compressors, termed as VAE-1 (Cheng et al., 2020) and VAE-2 (Ballé et al., 2018) respectively, for image compression.

**Regeneration Attack.** Following a recent benchmark (An et al., 2024), we utilize a diffusion model (Dhariwal & Nichol, 2021) pre-trained on ImageNet to perform regeneration attack. Specifically, the watermarked image will undergo multiple cycles of noising and be denoised through this pre-trained diffusion model for regeneration.

**Visually-Aware Attack.** Recently, Kassis et al. proposed the first practical universal attack, UnMarker (Kassis & Hengartner, 2024), on defensive watermarking. They introduce a visually-aware filter to attack semantic watermarks, such

as Tree-Ring and Gaussian Shading. Semantic watermarks only preserve the semantics of images, without regard for changes at the pixel level. GaussMarker also belongs to semantic watermarks.

## C. Metrics Details

**Bit Accuracy.** Assuming an $l$-bit binary watermark $w \in \{0,1\}^l$ is injected into the LDM, and the bit extracted from the generated image is $\tilde{\omega}$, the bit accuracy $\text{Acc}(w, \tilde{\omega}) \in (0,1)$ is defined as the ratio of the number of matching bits between $\omega$ and $\tilde{\omega}$ to $l$.

**TPR with Fixed FPR.** We need to predefine a threshold value $\tau \in \mathbb{R}$. If the detection score meets or exceeds the threshold $\tau$, it is concluded that the image indeed contains the watermark. To calculate the TPR with fixed FPR, e.g. TPR1%FPR, a natural way is finding a $\tau$ satisfying the given FPR, which can be used to obtain the TPR. However, if the controlled FPR is too low or the expected number of test samples is too large, we need to use LDM to generate many images, which increases the cost of evaluation. Previous research (Yu et al., 2021) proposes a more efficient approach to calculating this metric. They assume that each bit $\tilde{\omega}_i$ of the watermark extracted from un-watermarked images is uniformly distributed, e.g. $\tilde{\omega}_i \sim \text{Bernoulli}(0.5)$, where $\text{Bernoulli}(0.5)$ represents the Bernoulli distribution with a success probability of 0.5. Then, the FPR is defined as

$$\text{FPR}(\tau) = \sum_{i=\tau+1}^{l} \binom{l}{i} \frac{1}{2^l} = B_{\frac{1}{2}}(\tau+1, l-\tau), \quad (16)$$

where $B_{\frac{1}{2}}(\tau+1, l-\tau)$ is the regularized incomplete beta function. For $N$-users identification task, we multiply it by $N$ to obtain the expected FPR.

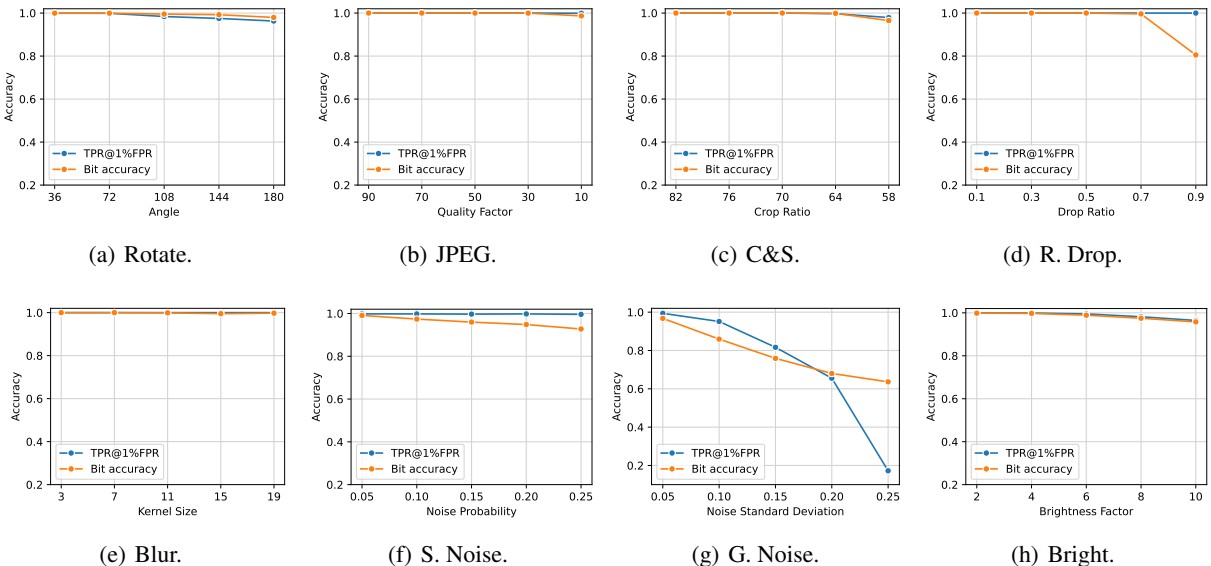

*Figure 5.* Detection performance of GaussMarker under eight image distortions of different intensities.

# D. More results

## D.1. Robustness

To further test the robustness, we conduct experiments using different intensities of noises for image distortions. Except for Gaussian Noise, TPR1%FPR and bit accuracy of Gauss-Marker remains above 90% and 80% respectively. Although the detection performance declines significantly with higher intensities of Gaussian noise, the quality of images is degraded too much (the second-to-last row of Fig.

## D.2. Dual-Domain and GNR

Tab. 8 presents the performance of five different versions of GaussMarker under various image distortions. The results show that each of the dual-domain watermarks has its advantages in terms of robustness. For example, the spatial-domain watermark obtain 19% and 56% higher TPR1%FPR than frequency-domain watermarks under JPEG and Random Drop respectively, while frequency-domain watermarks achieve 68% and 87% higher TPR1%FPR than spatial-domain watermarks under Rotate and C&S respectively. Injecting two types of watermarks simultaneously and fusing two detection scores leverages their respective strengths. However, the detection performance under Rotate and C&S is still not satisfactory, especially for the bit accuracy. Incorporating GNR significantly enhances the robustness against such two attacks, validating the design purpose of GNR.

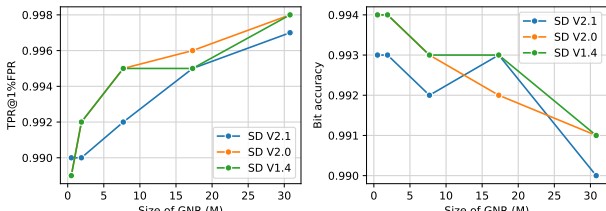

*Figure 6.* TPR1%FPR and bit accuracy of GaussMarker with different sizes of GNR across three versions of SD.

## D.3. Model Size of GNR

We test five different sizes of GNR (0.5M, 1.9M, 7.7M, 17.3M, and 30.8M), implementing each through setting the base feature dimension of the UNet to 16, 32, 64, 96, and 128 respectively. Fig. 6 shows the results. GaussMarker with a larger GNR can obtain better TPR1%FPR. Because a larger GNR can learn the invariance better to restore the Gaussian noise. However, GaussMarker with a smaller GNR can obtain higher bit accuracy. We consider the reason is that, when GNR can not learn the invariance perfectly, it is more likely for GNR to output the watermarked signal map more often, even if the input signal map does not contain our watermark. Therefore, small GNR can obtain higher bit accuracy but a little lower TPR1%FPR.

## D.4. Types of Fuser

In our main experiments, we implement the Fuser as a 2-layer MLP. We further test five other classifiers, 5-Nearest Neighborhood classifier with (KNN-5), SVM with a linear

*Table 8.* Ablation study on different modules of GaussMarker across three SD V1.4 / V2.0 / V 2.1 under various image distortions.

| Spatial | Freq. | GNR | DM | Clean | Rotate | JPEG | C&S | R. Drop | Blur | S. Noise | G. Noise | Bright | Average |
|---|---|---|---|---|---|---|---|---|---|---|---|---|---|
| ✓ | | | | 1.000 / 1.000 | 0.028 / 0.512 | 0.997 / 0.981 | 0.101 / 0.540 | 1.000 / 0.963 | 1.000 / 0.999 | 0.999 / 0.912 | 0.998 / 0.934 | 0.996 / 0.968 | 0.791 / 0.868 |
| | ✓ | | | 1.000 / - | 0.701 / - | 0.806 / - | 0.978 / - | 0.442 / - | 0.989 / - | 0.922 / - | 0.460 / - | 0.632 / - | 0.770 / - |
| ✓ | ✓ | | SD V2.1 | 1.000 / 1.000 | 0.652 / 0.512 | 0.996 / 0.981 | 0.978 / 0.540 | 0.998 / 0.963 | 1.000 / 0.999 | 1.000 / 0.912 | 0.924 / 0.934 | 0.968 / 0.968 | 0.946 / 0.868 |
| ✓ | | ✓ | | 1.000 / 1.000 | 0.996 / 0.998 | 0.989 / 0.997 | 1.000 / 1.000 | 1.000 / 0.963 | 1.000 / 1.000 | 0.000 / 0.991 | 0.899 / 0.968 | 0.994 / 0.989 | 0.875 / 0.990 |
| ✓ | ✓ | ✓ | | 1.000 / 1.000 | 0.997 / 0.998 | 0.996 / 0.997 | 1.000 / 1.000 | 1.000 / 0.963 | 1.000 / 1.000 | 0.999 / 0.991 | 0.989 / 0.968 | 0.993 / 0.989 | 0.997 / 0.990 |
| ✓ | | | | 1.000 / 1.000 | 0.009 / 0.515 | 0.998 / 0.985 | 0.140 / 0.541 | 1.000 / 0.969 | 1.000 / 0.999 | 1.000 / 0.923 | 0.998 / 0.939 | 0.999 / 0.977 | 0.794 / 0.872 |
| | ✓ | | | 1.000 / - | 0.768 / - | 0.838 / - | 0.974 / - | 0.397 / - | 0.989 / - | 0.943 / - | 0.547 / - | 0.688 / - | 0.794 / - |
| ✓ | ✓ | | SD V2.0 | 1.000 / 1.000 | 0.688 / 0.515 | 0.997 / 0.985 | 0.984 / 0.541 | 0.999 / 0.969 | 1.000 / 0.999 | 0.997 / 0.923 | 0.960 / 0.939 | 0.983 / 0.977 | 0.956 / 0.872 |
| ✓ | | ✓ | | 1.000 / 1.000 | 0.997 / 0.999 | 0.996 / 0.998 | 0.999 / 1.000 | 1.000 / 0.969 | 1.000 / 1.000 | 0.962 / 0.992 | 0.907 / 0.970 | 0.997 / 0.993 | 0.984 / 0.991 |
| ✓ | ✓ | ✓ | | 1.000 / 1.000 | 0.997 / 0.999 | 0.998 / 0.998 | 1.000 / 1.000 | 1.000 / 0.969 | 1.000 / 1.000 | 0.996 / 0.992 | 0.993 / 0.970 | 0.998 / 0.993 | 0.998 / 0.991 |
| ✓ | | | | 1.000 / 1.000 | 0.006 / 0.517 | 0.998 / 0.986 | 0.078 / 0.538 | 1.000 / 0.970 | 1.000 / 0.999 | 1.000 / 0.915 | 0.996 / 0.937 | 0.998 / 0.979 | 0.786 / 0.871 |
| | ✓ | | | 1.000 / - | 0.586 / - | 0.866 / - | 0.977 / - | 0.458 / - | 0.989 / - | 0.896 / - | 0.438 / - | 0.783 / - | 0.777 / - |
| ✓ | ✓ | | SD V1.4 | 1.000 / 1.000 | 0.576 / 0.517 | 0.998 / 0.986 | 0.983 / 0.538 | 1.000 / 0.970 | 1.000 / 0.999 | 0.997 / 0.915 | 0.944 / 0.937 | 0.987 / 0.979 | 0.943 / 0.871 |
| ✓ | | ✓ | | 1.000 / 1.000 | 0.997 / 0.999 | 0.997 / 0.998 | 0.999 / 0.999 | 1.000 / 0.970 | 1.000 / 1.000 | 0.997 / 0.991 | 0.980 / 0.965 | 0.996 / 0.994 | 0.996 / 0.991 |
| ✓ | ✓ | ✓ | | 1.000 / 1.000 | 0.998 / 0.999 | 0.998 / 0.998 | 0.999 / 0.999 | 1.000 / 0.970 | 1.000 / 1.000 | 1.000 / 0.991 | 0.988 / 0.965 | 0.996 / 0.994 | 0.998 / 0.991 |

*Table 9.* Results of different Fusers across three SD V1.4 / V2.0 / V 2.1.

| Fuser Type | TPR1%FPR | ROC-AUC |
|---|---|---|
| KNN-5 | 0.997 / 0.994 / 0.993 | 0.998 / 0.997 / 0.996 |
| Linear SVM | 0.998 / 0.991 / 0.995 | 0.999 / 0.999 / 0.999 |
| RBF SVM | 0.998 / 0.992 / 0.996 | 0.999 / 0.999 / 0.999 |
| Random Forest | 0.998 / 0.993 / 0.994 | 0.999 / 0.999 / 0.999 |
| Decision Tree | 0.661 / 0.647 / 0.875 | 0.988 / 0.982 / 0.988 |
| MLP | 0.998 / 0.998 / 0.997 | 0.999 / 0.999 / 0.999 |

kernel (Linear SVM), SVM with a radial basis function kernel (RBF SVM), Random Forest with 100 trees, and Decision Tree with Gini impurity. Since the Fuser does not affect the bit accuracy, we report another detection metric, ROC-AUC, following previous works (Wen et al., 2023; Ci et al., 2024). As shown in Tab. 9, except for Decision Tree, all types of Fuser achieve over 99% TPR1%FPR and ROC-AUC, and MLP obtains the highest average TPR1%FPR.

### D.5. Effects of the Mask Radius

In our main experiments, the mask radius $r_f$ is set as 4. We further test five other values, 0, 2, 6, 8, and 10. Fig. 7 is the visualization of the masks with different $r_f$. Fig. 8 presents the detection performance of GaussMarker with different $r_f$. When $r_f = 0$, e.g. no frequency watermark, the TPR1%FPR of GaussMarker falls below 90%, indicating the importance of injecting frequency-domain watermarks. However, when $r_f$ gets too large, the TPR1%FPR starts to decrease. This is because that, too large $r_f$ will change the signal map of $z_T^s$ a lot. Since changing the signal map of $z_T^s$ means changing the injected watermark, the detection score of spatial-domain watermark will get inaccurate, as shown in the right part of Fig. 8.

### D.6. Sampling Steps

We explore how the number of sampling steps during generation and DDIM inversion affects the detection performance of GaussMarker. As present in Tab. 10, both larger sam-

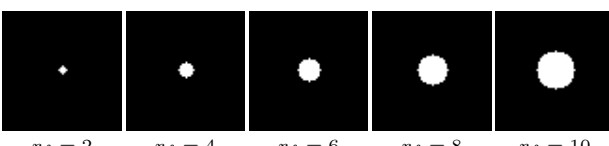

| $r_f = 2$ | $r_f = 4$ | $r_f = 6$ | $r_f = 8$ | $r_f = 10$ |

*Figure 7.* Visualization of the masks with different $r_f$ for frequency-domain watermarks.

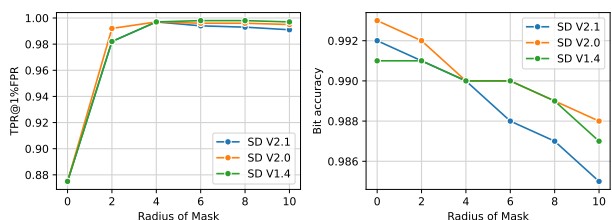

*Figure 8.* TPR1%FPR and bit accuracy of GaussMarker when using masks with different radius $r_f$ for frequency-domain watermark. $r_f = 0$ means GaussMarker does not inject the frequency-domain watermark.

*Table 10.* Effects of sampling steps on TPR1%FPR / bit accuracy of GaussMarker using SD V2.1.

| Gen. Steps | Inv. Steps | | | |
|---|---|---|---|---|
| | 10 | 25 | 50 | 100 |
| 10 | 0.993 / 0.980 | 0.994 / 0.983 | 0.994 / 0.983 | 0.995 / 0.984 |
| 25 | 0.996 / 0.986 | 0.996 / 0.989 | 0.998 / 0.990 | 0.998 / 0.990 |
| 50 | 0.996 / 0.987 | 0.997 / 0.989 | 0.997 / 0.990 | 0.998 / 0.990 |
| 100 | 0.996 / 0.987 | 0.996 / 0.989 | 0.997 / 0.990 | 0.997 / 0.990 |

pling steps during generation and inversion improve the TPR1%FPR and bit accuracy of GaussMarker. However, even when both generation steps and inversion steps are only 10, the TPR1%FPR and bit accuracy of GaussMarker still remain above 99% and 98% respectively.

