# OpenReview forum: "GaussMarker: Robust Dual-Domain Watermark for Diffusion Models"
_ICML.cc/2025/Conference — ICML 2025 poster_

### Official Review · Reviewer_52ML · 2025-03-12

**Overall Recommendation:** 3

**Summary:**

This paper introduces GaussMarker, the first dual-domain watermarking technique tailored for diffusion models. The authors preoposed a
Novel Dual-Domain Watermarking taht is designed for diffusion models without requiring any fine-tuning, while still achieving strong robustness. The authors also develop GNR, which is trained independently from the diffusion models. This component notably enhances the watermark's resilience against rotation and cropping attacks. Extensive experiments across three stable diffusion models and eight types of image distortions demonstrate that GaussMarker outperforms existing methods in terms of true positive rate and bit accuracy.

**Claims And Evidence:**

The claims are supported by experiemnts

**Essential References Not Discussed:**

N/A

**Experimental Designs Or Analyses:**

Ablation experiments are conducted, various attacking method are considered when evaluating the method.

However, in table 1, it seems that the propsoed method are more rebust than other watermarking method only under rotaion and C&S attack. Any comments?
Moreover, as the GNR is trained in latent space, does it mean that it would be reobust on image space attack like JPEG and Brightness.

**Methods And Evaluation Criteria:**

The method includes two part: the first is a dual-domain watermark encoder and the second is a GNR for watermark robustness improvement. However, the assumptions that $\left(\mathcal{T}\left(x^{s, f}\right)\right) \approx \mathcal{T}\left(z_T^{s, f}\right)$ might not stand in some cases where the watermarked images have some differences with the source image.

**Other Comments Or Suggestions:**

Comparing other attacks, why applying Gaussian Noise attack in a multiple user applicaiton would grealty downgrade the accuracy is not clear, could the author provide some comment on this?

**Other Strengths And Weaknesses:**

Strengths:

The dual domain design and score fusing is novel and can improve the robustness for watermark detection.

Weaknesses:

Even though the proposed method does not achieve state-of-the-art performance under all attack, it achieves competitive results.
However, comparing with other methods with similiar performance (RingID, TreeRing, etc), the proposed method needs additonal training.

**Questions For Authors:**

N/A

**Relation To Broader Scientific Literature:**

N/A

**Theoretical Claims:**

The theoretical claims are correct

---

> ### Author Rebuttal · Authors · 2025-03-31
>
> **Q1:** However, in table 1, it seems that the propsoed method are more rebust than other watermarking method only under rotaion and C\&S attack. Any comments? Moreover, as the GNR is trained in latent space, does it mean that it would be reobust on image space attack like JPEG and Brightness.
>
> **A1.1:** GaussMarker also achieves better detection performance than Stable Signature, TreeRing and PRC under S. Noise, G. Noise, and Bright attacks, as present in Table 1. A major shortcoming of existing methods is that they do not perform well across different image distortions, whereas GaussMarker exhibits relatively consistent robustness across various image distortions.
>
> **A1.2:** Yes, GNR can inherit the robustness of LDM, as present in the 1st and 4th rows of Table 6. However, since GNR uses an approximate objective, it may extract a little high detection score from an unwatermarked image sometimes (low TPR@1%FPR but high bit accuracy in the 4th row and S. Noise line of Table 6). Therefore, it is necessary to fuse the frequency score for a lower FPR.
>
> **Q2:** Even though the proposed method does not achieve state-of-the-art performance under all attack, it achieves competitive results. However, comparing with other methods with similiar performance (RingID, TreeRing, etc), the proposed method needs additonal training.
>
> **A2:** GaussMarker does need additional training, but the cost is minimal and the training is model-agnostic. As presented in the table below, the training of GNR only needs 72 minutes on 1 V100 32G GPU. Moreover, the time overhead incurred by GNR and Fuser during the detection phase is almost negligible.
>
> | Phase     | Inversion | GNR         | Fuser          |
> |-----------|-----------|-------------|----------------|
> | Training  | -         | 72 min      | 1.4×10⁻¹ s     |
> | Detection | 6.5 s     | 1.2×10⁻³ s  | 1.0×10⁻⁴ s     |
>
> **Q3:** Comparing other attacks, why applying Gaussian Noise attack in a multiple user applicaiton would grealty downgrade the accuracy is not clear, could the author provide some comment on this?
>
> **A3:** When the number of users increases, higher bit accuracy is required in extracting watermarks from images to ensure high identification accuracy. For example, consider injecting a 3-bit watermark (ignoring unwatermarked images for simplicity). With two users assigned watermarks \{0,0,0\} and \{1,1,1\}, an estimated watermark $\tilde{w}$=\{0,0,1\} allows us to correctly identify the first user with just 66.7\% bit accuracy.
>
> As presented in Table 1, the bit accuracy under Gaussian Noise is the most unstable. Under Gaussian Noise attack, GaussMarker only obtains 0.989 TPR\@1\%FPR, which means that nearly 11 watermarked images don't get high bit accuracy. Therefore, it is affected by the number of users greatly.

---

### Official Review · Reviewer_4dSB · 2025-03-12

**Overall Recommendation:** 3

**Summary:**

This paper presents **GaussMarker**, a novel watermarking method for diffusion models, with the following key contributions:
1. **Dual-Domain Watermarking**: Embeds watermarks in both the spatial and frequency domains of Gaussian noise to enhance robustness.
2. **Gaussian Noise Restorer (GNR)**: A model-independent learnable module that improves watermark detection robustness, especially against geometric attacks like rotation and cropping.
3. **Score Fusion**: Combines detection signals from spatial and frequency domains to improve extraction accuracy.

Experimental results demonstrate that **GaussMarker** outperforms existing methods across three versions of Stable Diffusion (V1.4, V2.0, V2.1) under eight common distortions (e.g., rotation, blur, noise, JPEG compression) and four advanced attacks (e.g., VAE compression, diffusion model regeneration, UnMarker attack).

**Claims And Evidence:**

The paper makes the following main claims:
1. Existing single-domain watermarking methods lack robustness against geometric attacks (e.g., rotation and cropping), while dual-domain watermarking significantly improves detection performance.
2. **GNR** enhances watermark robustness by restoring corrupted watermark information.
3. Score fusion improves watermark detection accuracy compared to using single-domain detection alone.

While the paper provides experimental support for these claims, several issues remain:
- **Effectiveness of dual-domain watermarking**: The paper shows that combining spatial and frequency-domain watermarks improves detection accuracy (Table 4), but lacks theoretical analysis explaining why these two approaches complement each other.
- **Role of GNR**: The study (Figure 2) demonstrates that GNR improves robustness against rotation and cropping, and Table 4 shows increased detection accuracy. However, the learning objective of GNR (Equations 9 and 10) is based on noise signal recovery, without considering more complex image transformations, which might limit its generalization. Additionally, GNR is only optimized for rotation and cropping but its effectiveness against other attacks (e.g., Gaussian noise, blurring) remains unclear.

**Essential References Not Discussed:**

Most of the essential references are discussed.

**Experimental Designs Or Analyses:**

The experimental design is generally solid, but there are areas for improvement:
1. **Limitations of GNR**:
   - As mentioned in **Theoretical Claims**

2. **Effectiveness of score fusion**:
   - The correlation between spatial and frequency-domain scores is not studied.

**Methods And Evaluation Criteria:**

The paper employs reasonable evaluation methods:
- **Baselines**: Compares against various state-of-the-art watermarking methods, including **Tree-Ring, Gaussian Shading, PRC, LatentTracer** (all tuning-free) and **Stable Signature** (a tuning-based method).
- **Datasets**: Evaluates on **Stable Diffusion V1.4, V2.0, V2.1** using **MS-COCO** to generate 512×512 watermarked images.
- **Metrics**:
  - **TPR@1%FPR**: True positive rate at 1% false positive rate, measuring detection performance.
  - **Bit Accuracy**: Measures the correctness of extracted watermark information.
  - **FID** and **CLIP-Score**: Assess image quality.

**Other Comments Or Suggestions:**

See Weaknesses.

**Other Strengths And Weaknesses:**

**Strengths**

- **Innovative dual-domain watermarking**: The proposed *GaussMarker* introduces a novel approach by embedding watermarks in both the spatial and frequency domains. This idea is well-motivated and aligns with the intuition that leveraging multiple domains can enhance robustness.
- **No fine-tuning required**: Unlike tuning-based watermarking methods, *GaussMarker* does not require modifying diffusion model parameters, making it computationally efficient and more practical for real-world deployment.
- **Comprehensive experiments**: The paper evaluates *GaussMarker* on multiple versions of Stable Diffusion (V1.4, V2.0, V2.1) and benchmarks against various state-of-the-art watermarking techniques. It tests robustness under eight image distortions and four advanced attacks, demonstrating superior performance.
- **Gaussian Noise Restorer (GNR) for improved robustness**: The introduction of GNR helps mitigate the impact of geometric transformations (rotation, cropping), significantly improving detection accuracy in such cases.

**Weaknesses**

- **Validity of GNR’s Learning Objective:**
The paper assumes that the Gaussian Noise Restorer (GNR) can effectively restore watermarked noise under transformations like rotation and cropping (Equation 10). However, it does not provide formal proof of this claim. Furthermore, while **Tree-Ring** mentions that watermark signals exhibit invariance under geometric transformations, it does not extensively discuss robustness against **non-linear transformations** such as **JPEG compression and blurring**.

This raises several critical questions:
1. **Scope of Equation 10**: What specific types of transformations are covered by Equation 10? Does it include JPEG compression and blurring?
   - If **JPEG and blurring** are included, can the authors provide more rigorous theoretical justification or additional experiments to support this claim?
   - If **JPEG and blurring** are not included, the authors should explicitly state the limitations of Equation 10 and clarify the scope of transformations where GNR is applicable.

2. **Effectiveness of GNR beyond rotation and cropping**:
   - In **Table 6**, the ablation study suggests that GNR improves watermark robustness across **all distortions**, not just rotation and cropping.
   - However, the training transformation **T** only includes **rotation and cropping**, which raises concerns about potential overfitting to these specific noise layers.
   - Typically, deep learning models trained on a limited set of distortions **tend to overfit** to those distortions while exhibiting **weaker generalization** to unseen noise types.
   - If GNR was only trained on **rotation and cropping**, why does it still improve robustness against **other distortions** (e.g., JPEG compression, blurring)?
   - Can the authors provide a **rational explanation** for this unexpected gain?

**Questions For Authors:**

1. **Scope of Equation 10**: Equation 10 is designed to improve robustness against rotation and cropping. Does it also apply to **non-geometric transformations** like JPEG compression and blurring? If so, can you provide a formal explanation or additional experiments? If not, can you clarify its limitations?
2. **Generalization of GNR**: In Table 6, GNR consistently improves detection across all distortions, despite being trained only on rotation and cropping. Given that deep-learning models often overfit to specific noise patterns, how do you explain this unexpected gain?
3. **Complementarity of spatial and frequency-domain watermarks**: The paper claims that dual-domain embedding improves robustness. Can you provide an analysis or empirical study to justify why spatial and frequency-domain watermarks are complementary?

**Relation To Broader Scientific Literature:**

The paper situates itself well within the watermarking literature, comparing tuning-free and tuning-based methods.

**Theoretical Claims:**

The paper’s main theoretical contribution is the introduction of **GNR** for watermark restoration. However, several theoretical gaps remain:

**Validity of GNR’s Learning Objective:**
The paper assumes that the Gaussian Noise Restorer (GNR) can effectively restore watermarked noise under transformations like rotation and cropping (Equation 10). However, it does not provide formal proof of this claim. Furthermore, while **Tree-Ring** mentions that watermark signals exhibit invariance under geometric transformations, it does not extensively discuss robustness against **non-linear transformations** such as **JPEG compression and blurring**.

This raises several critical questions:
1. **Scope of Equation 10**: What specific types of transformations are covered by Equation 10? Does it include JPEG compression and blurring?
   - If **JPEG and blurring** are included, can the authors provide more rigorous theoretical justification or additional experiments to support this claim?
   - If **JPEG and blurring** are not included, the authors should explicitly state the limitations of Equation 10 and clarify the scope of transformations where GNR is applicable.

2. **Effectiveness of GNR beyond rotation and cropping**:
   - In **Table 6**, the ablation study suggests that GNR improves watermark robustness across **all distortions**, not just rotation and cropping.
   - However, the training transformation **T** only includes **rotation and cropping**, which raises concerns about potential overfitting to these specific noise layers.
   - Typically, deep learning models trained on a limited set of distortions **tend to overfit** to those distortions while exhibiting **weaker generalization** to unseen noise types.
   - If GNR was only trained on **rotation and cropping**, why does it still improve robustness against **other distortions** (e.g., JPEG compression, blurring)?
   - Can the authors provide a **rational explanation** for this unexpected gain?

---

> ### Author Rebuttal · Authors · 2025-03-31
>
> **Q1:** Scope of Equation 10: Equation 10 is designed to improve robustness against rotation and cropping. Does it also apply to non-geometric transformations like JPEG compression and blurring? If so, can you provide a formal explanation or additional experiments? If not, can you clarify its limitations?
>
> **A1:** Equation 10 does not apply to common non-geometric transformations in our experiments. We will add this limitation. However, during detection, GNR takes the signal map of Gaussian noise that is estimated by LDM as input. Therefore, GNR can inherit the robustness of LDM (the 1st row of Table 6).
>
> **Q2:** Generalization of GNR: In Table 6, GNR consistently improves detection across all distortions, despite being trained only on rotation and cropping. Given that deep-learning models often overfit to specific noise patterns, how do you explain this unexpected gain?
>
> **A2:** Since the objective of GNR is approximate, as present in Figure 2 (nearly 30%-42%), we use random sign flipping (p=0.35) for GNR training. This prevents GNR from overfitting and ensures its good generalization. Therefore, GNR also enhances the robustness of GaussMarker under other attacks to some extent.
>
> We provide additional ablation results ( TPR@1%FPR / bit acc. on SD V2.1) in the table below to verify this. With random sign flipping, the average TPR@1%FPR and bit accuracy of GaussMarker increases by 5.7% and 12.1%, respectively. Although GaussMarker gains the main improvement under Rotate and C\&S attacks, the improvement under S. Noise, G. Noise, and Bright is also significant, especially on the bit accuracy.
>
> | Methods            | Clean       | Rotate      | JPEG        | C\&S        | R. Drop     | Blur        | S. Noise    | G. Noise    | Bright      | Average     |
> |--------------------|-------------|-------------|-------------|-------------|-------------|-------------|-------------|-------------|-------------|-------------|
> | GaussMarker w/o flipping | 1.000 / 1.000 | 0.587 / 0.513 | 0.990 / 0.982 | 0.980 / 0.540 | 1.000 / 0.963 | 1.000 / 0.999 | 0.999 / 0.912 | 0.988 / 0.934 | 0.916 / 0.967 | 0.940 / 0.868 |
> | GaussMarker w/ flipping  | 1.000 / 1.000 | 0.997 / 0.998 | 0.996 / 0.997 | 1.000 / 1.000 | 1.000 / 0.963 | 1.000 / 1.000 | 0.999 / 0.991 | 0.989 / 0.968 | 0.993 / 0.989 | 0.997 / 0.990 |
>
> **Q3:** Complementarity of spatial and frequency-domain watermarks: The paper claims that dual-domain embedding improves robustness. Can you provide an analysis or empirical study to justify why spatial and frequency-domain watermarks are complementary?
>
> **A3:** We believe that the effectiveness of combining is built on the theoretical foundation of ensemble learning [a]. When both frequency-domain watermarking and spatial-domain watermarking can achieve precise and diverse detection results, as present in Table 6, fusing them may lead to more robust detection performance. This has been fully demonstrated by our experimental results.
>
> [a] https://jmlr.org/papers/v24/23-0041.html

---

### Official Review · Reviewer_u9TC · 2025-03-13

**Overall Recommendation:** 3

**Summary:**

This paper proposed **GaussMarker**, which embeds watermarks into the noise vector of diffusion models within both the spatial and frequency domains. To enhance the detection robustness of watermarks, the authors propose a learnable Gaussian Noise Restorer (GNR) that is capable of restoring from the distorted noise vector, especially under the rotation and cropping attacks. Experiments show that **GaussMarker** is robust against most common image transformations and four advanced watermarking removal attacks.

**Claims And Evidence:**

Yes.

**Essential References Not Discussed:**

No.

**Experimental Designs Or Analyses:**

Yes, most of the experimental designs are valid and make sense. Below, I draw several questions and confusion from the paper.

1. Why does **GaussMarker** largely outperform other methods when evaluated against the regeneration attack? Can you explain the underlying mechanism?
2. How do you evaluate the identification accuracy with multiple users? Can you explain the experimental details?
3. For rotation attacks, is the GNR capable of generalizing to other useen rotation degrees (angles not used during training the GNR)? My concern is that since the CNN-based networks face challenges in handling the image rotations, how does GNR overcome this?

**Methods And Evaluation Criteria:**

This paper compares several updated watermarking strategies, using standard metrics for detection (TPR and Bit Accuracy) and quality evaluation (FID and CLIP-score) and widely used prompt sets from MS-COCO for image generation. All of them make sense.

**Other Comments Or Suggestions:**

The method description needs further proofreading to make the paper more fluent and easy to follow.

**Other Strengths And Weaknesses:**

**Strengths**:
1. The proposed **GaussMarker** is robust than most advanced baselines, making it an effective training-free watermarking strategy.
2. The motivation is clear and straightforward.

**Weaknesses**:
1. The formulation and method description of this paper needs further proofreading and justification. For instance, $s$ is used to represent both signal map and spatial space, and sometimes they are mixed $s^{s,f}$; $s^{s,f}$ is less rigorously defined as signal map is only introduced in the **s**patial space but not in the **f**requency space (please point out my mistakes if I misunderstand the method).
2. It is confusing whether GNR requires retraining for each signal map. If so, the scalability of GNR is reduced, and how to select which GNR to use when multiple users generate images from the same LDM?
3. The idea of GNR is similar to the robustly trained watermark decoder (in Stable Signature) and is only designed to enhance the robustness of rotation and cropping.

**Questions For Authors:**

1. How do you derive the Equation in lines 222-224 (needs index) from Equation 10?
2. In table 4, why does SD V 2.1's TPR performance get a significant drop compared to SD V 2.0 in the last two rows (from 0.997 -> 0.875), while the bit ACC remains the same? It seems that Spatial + GNR is sufficient to get a robust watermarking performance.

**Relation To Broader Scientific Literature:**

**GaussMarker** can be used to protect the intellectual property of diffusion-based generative models and safeguard their generated images from potential misuse.

**Theoretical Claims:**

No theoretical claims in this paper.

---

> ### Author Rebuttal · Authors · 2025-03-31
>
> **Q1:** Why does GaussMarker largely outperform other methods when evaluated against the regeneration attack?
>
> **A1:** This advantage stems from GNR, as shown in the SD V2.1 results presented in the table below. Regeneration using diffusion models mainly involves semantic editing, which may also include spatial-like transformations. Notably, since GNR is trained using the composition of rotation and cropping with random sign flipping (p=0.35) for preventing overfitting, it can learn the invariance to various spatial-like transformations. Therefore, GNR enhances GaussMarker's ability to resist spatial-like transformations introduced by regeneration attacks.
>
> | Spatial | Frequency | GNR | TPR@1%FPR | Bit Acc. |
> |---------|-----------|-----|-----------|----------|
> | ✓       |           |     | 0.008     | 0.524    |
> |         | ✓         |     | 0.295     | -        |
> | ✓       | ✓         |     | 0.345     | 0.524    |
> | ✓       |           | ✓   | 0.660     | 0.865    |
> | ✓       | ✓         | ✓   | 0.667     | 0.865    |
>
> **Q2:** How do you evaluate the identification accuracy with multiple users?
>
> **A2:** We adopt an approximate algorithm as detailed in Appendix C, following Gaussian Shading. Specifically, we first compute a theoretical threshold $\tau$ based on the predetermined number of users, $N$ and expected FPR. Given a watermarked image, if its bit accuracy exceeds this threshold, we consider the user generating it will be correctly identified among the $N$ users. A practical multi-user identification extension of GaussMarker is provided in Response A5.
>
> **Q3:** For rotation attacks, is the GNR capable of generalizing to other useen rotation degrees (angles not used during training the GNR)? My concern is that since the CNN-based networks face challenges in handling the image rotations, how does GNR overcome this?
>
> **A3:** GNR is trained with dense rotation angles randomly sampled from (-180$^{\circ}$, 180$^{\circ}$), and can converge easily. Therefore, even with unseen rotation degrees, they are likely to be approximated with minor errors by nearby angles.
>
> **Q4:** The formulation and method description of this paper needs further proofreading and justification.
>
> **A4:** We will improve the formulation and method description.
>
> **Q5:** It is confusing whether GNR requires retraining for each signal map. If so, the scalability of GNR is reduced, and how to select which GNR to use when multiple users generate images from the same LDM?
>
> **A5:** When we have multiple users, we also only need to train one GNR. Each LDM only has one unique model-watermark $w \in$ {0,1}$^l$ (or signal map) with its corresponding GNR. Each user will be assigned a unique key $k \in$ {0,1}$^l$ and a unique user-watermark $w_u \in$ {0,1}$^l$ with $w_u = \text{XOR}(w,k)$. We use the estimated $\tilde{w}$ to estimate the user-watermark $\tilde{w}_u = \text{XOR}(\tilde{w},k)$ for calculating the bit accuracy. This easy extension is similar to Gaussian shading, so we omit that detail from the paper. We can add it if needed.
>
> **Q6:** The idea of GNR is similar to the robustly trained watermark decoder (in Stable Signature) and is only designed to enhance the robustness of rotation and cropping.
>
> **A6:** The core ideas are similar, but training GNR is much more efficient. GNR does not require additional datasets for training, while Stable Signature needs to train its watermark decoder and VAE on the COCO dataset. Besides, in addition to rotation and cropping, GNR also includes random sign flipping and is robust to many spatial-like transformations in the pixel space, as detailed in Response A1.
>
> **Q7:** How do you derive the Equation in lines 222-224 (needs index) from Equation 10?
>
> **A7:** In Equation 10, GNR takes Gaussian noise as input and outputs its restored version. However, as shown in Figure 2, we find that the approximate invariance exists only in the signal space. Therefore, we redefine GNR to take the signal map of Gaussian noise as input and output the restored signal map, as formalized in the Equation in lines 222-224.
>
> **Q8:** In table 4, why does SD V 2.1's TPR performance get a significant drop compared to SD V 2.0 in the last two rows (from 0.997 $\rightarrow$ 0.875), while the bit ACC remains the same? It seems that Spatial + GNR is sufficient to get a robust watermarking performance.
>
> **A8:** This is because the Spatial+GNR sometimes extracts high bit accuracy from unwatermarked images, which lowers the FPR and TPR@1%FPR, especially on stronger models (SD V2.1 is a further fine-tuned version of SD V2.0). Note that bit accuracy is only calculated from watermarked images, so it is less affected. The frequency score helps to reduce the FPR through score fusion.

---

### Official Review · Reviewer_YNzZ · 2025-03-17

**Overall Recommendation:** 3

**Summary:**

The paper introduces GaussMarker, a novel semantic watermark technique based on diffusion models. Different from previous works, GaussMarker adds watermarks in both the pixel and frequency domain of images. During detection, GaussMarker trains two additional components: 1. Gaussian noise restorer (GNR) for restoring the geometric attacks and 2. Fuser for fusing the detection score between pixel space and frequency space detection. Experiment results demonstrate the superiority of GaussMarker towards geometric attacks (Rotate, Crop&Scale).

**Claims And Evidence:**

The claim "GaussMarker is tunning-free watermarks" is improper. Although GaussMarker doesn't need to tune the diffusion model, it still needs to train the Gaussian noise restorer and Fuser. Thus it is unfair to categorize GaussMarker as tunning-free watermarks.

**Essential References Not Discussed:**

The paper has a thorough discussion of previous works.

**Experimental Designs Or Analyses:**

The paper experiments with the robustness and generation fidelity of their proposed method on three different versions of Stable Diffusion and various attacks. Furthermore, it contains ablation studies as well as experiments on multiple user cases. The experiments are thorough.

**Methods And Evaluation Criteria:**

Yes, the proposed method is make sense for the problem.

**Other Comments Or Suggestions:**

1. Some problems in Figure 3, we couldn't visualize the results such as rotate, and C&S. The author needs a better illustration.

**Other Strengths And Weaknesses:**

## Strength
1. By incorporating the two additional components (GNR and fuser) and adding watermarks in dual domains, GaussMarker achieves much better robustness. This contribution could be clearly illustrated in Table 4.

## Weakness
1. The introduction of additional components caused more concerns:

    a. More computation expensive: How much is the additional computation cost introduced by the GNR and the fuser for both training and detection?

    b. Effect on generation fidelity. As shown in Table 2, compared with the most direct baseline Tree-Ring, GaussMarker achieves slightly better FID and slightly worse CLIP Score. But what is the effect of spatial watermark and frequency watermark on the fidelity is still unclear.  Similar to Table 4, the paper should have an ablation study related to this.

    c. Generalizability of these two components: does the paper train three GNRs and users for three stable diffusions or one for all of them?

**Questions For Authors:**

1. For Table 4, compare with the 4th and 5th rows, including the frequency significantly improve TPR1%FPR for V2.1. But this frequency watermark is not important for V1.4 and V2.0. Could the author discuss the difference between different versions of stable diffusion and why the frequency watermark only matters for V2.1?
2. What's the relation between watermark capacity and the number of users in Figure 3? If the watermark capacity is 2^10, is the number of users 2^(2^10)?
3. In Figure 3, why increasing the watermark capacity will make the watermark less vulnerable to Random Drop? why increasing the number of users will make the watermark less vulnerable to Gaussian noise? Is there any discussion?

**Relation To Broader Scientific Literature:**

The key contribution of this paper, watermark, is related to AI safety, and copyright protection problems.

**Theoretical Claims:**

There are no theoretical claims in this paper.

---

> ### Author Rebuttal · Authors · 2025-03-31
>
> **Q1:** About "tunning-free".
>
> **A1:** We consider GaussMarker to be a tuning-free method. The term "tuning-free" implies that SDs cannot be fine-tuned due to computational costs and the watermarking method can be attached to the model in a plug-and-play manner without touching the model weights. GaussMarker avoids fine-tuning SDs and instead trains only two small modules, which require minimal computational resources, as elaborated in Response 7.
>
> **Q2:** Generalizability of these two components.
>
> **A2:** We train a GNR for all SDs due to its model-independent characteristics. We train a Fuser for each SD, as the training data for the Fuser is generated from the corresponding SD.
>
> **Q3:** Some problems in Figure 3.
>
> **A3:** Some curves overlap at y=1. We will improve the illustration.
>
> **Q4:** Why does frequency watermark only matter for V2.1?
>
> **A4:** The frequency watermark matters more for stronger SDs **when using GNR**. In our experiments, we find that GNR occasionally extracts a high bit accuracy from unwatermarked images, especially on stronger SDs. This results in the need for a higher threshold to maintain the expected 1\% FPR, which in turn lowers the TPR@1%FPR.
>
> For instance, in Table 6, GNR+Spatial demonstrates better bit accuracy but worse TPR@1%FPR compared to Spatial under Gaussian Noise and Salt-and-Pepper Noise on SD V2.1. This significant drop does not occur with SD V1.4 and V2.0. Given the stable FPR of the frequency watermark [a], the frequency score becomes crucial for achieving a lower FPR through score fusion, especially on SD V2.1.
>
> [a] https://arxiv.org/pdf/2305.20030
>
> **Q5:** The relation between watermark capacity and the number of users in Figure 3.
>
> **A5:** The watermark capacity and number of users in Figure 3 are not directly related. The watermark capacity, denoted by
> $l$, refers to the number of bits available for watermarking. The number of users that can be assigned unique watermarks must be much less than $2^l$, and a larger $l$ usually supports more users. In Figure 3(b), with $l=256$, we assess how accurately GaussMarker can identify the correct user as the number of users grows.
>
> **Q6:** In Figure 3, why increasing the watermark capacity will make the watermark less vulnerable to Random Drop? why increasing the number of users will make the watermark less vulnerable to Gaussian noise?
>
> **A6.1:** Maybe you want to use 'more' instead of 'less'. As shown in Equation 5, we use a voting strategy (average pooling) to estimate each bit in the watermark $\tilde{\omega} \in$ {0,1}$^l$ based on $cwh/l$ signs in the signal map $\tilde{s} \in$ {0,1}$^{cwh}$.  When $l$  increases, fewer signs can be used for voting, which means that sign estimation needs to be more accurate for estimating the $\tilde{\omega}$. Random Drop masks 80% of the image with black pixels, significantly impacting sign accuracy. Therefore, the bit accuracy under this attack is most affected by the watermark capacity.
>
> **A6.2:** When the number of users increases, higher bit accuracy is required in extracting watermarks from images to ensure high identification accuracy. For example, consider injecting a 3-bit watermark (ignoring unwatermarked images for simplicity). With two users assigned watermarks {0,0,0} and {1,1,1}, an estimated watermark $\tilde{w}$={0,0,1} allows us to correctly identify the first user with just 66.7% bit accuracy.
>
> As presented in Table 1, the bit accuracy under Gaussian Noise is the most unstable. Therefore, it is affected by the number of users most. Note that we set the watermark capacity to $l=256$ in this experiment, thus, the performance under Random Drop is great with its 1.000 TPR@1%FPR. However, under Gaussian Noise attack, GaussMarker only obtains 0.989 TPR@1%FPR, which means that nearly 11 watermarked images don't get high bit accuracy.
>
> **Q7:** Computation cost.
>
> **A7:** The cost is minimal, as presented in the table below. These experiments are conducted on 1 V100 32G GPU.
>
> | Phase     | Inversion | GNR         | Fuser          |
> |-----------|-----------|-------------|----------------|
> | Training  | -         | 72 min      | 1.4×10⁻¹ s     |
> | Detection | 6.5 s     | 1.2×10⁻³ s  | 1.0×10⁻⁴ s     |
>
> **Q8:** Ablation study on fidelity.
>
> **A8:** The ablation results are shown in the table below. Both spatial and frequency watermarks help preserve the original CLIP Score. For FID, the spatial watermark performs better than the frequency one. Combining both methods often requires more editing, leading to slightly worse CLIP Scores and FID for the dual-domain watermark compared to single-domain approaches. Improving the visual quality of watermarked images using GaussMarker could be a direction for future work.
>
> | Spatial | Frequency | Ave. CLIP Score↑  | Ave. FID ↓   |
> |---------|-----------|--------|--------|
> |         |           | 0.3567 | 24.89 |
> | ✓       |           | 0.3568 | 24.36 |
> |         | ✓         | 0.3568 | 24.78 |
> | ✓       | ✓         | 0.3545 | 24.85 |

---

### Decision · Program_Chairs · 2025-05-01

**Decision:**

Accept (poster)

**Comment:**

This paper has received all weak acceptances in the final recommendations. This paper proposed GaussMarker, which embeds watermarks into the noise vector of diffusion models within both the spatial and frequency domains. Although the reviewers expressed concerns about the unclear justification and insufficient experimental analysis, the authors' rebuttal effectively addressed several of these issues. This led to a consensus among the reviewers in favor of acceptance. Consequently, the Area Chair has decided to accept this paper.